# A stapled lipopeptide platform for preventing and treating highly pathogenic viruses of pandemic potential

Gregory H. Bird[1,2], J. J. Patten [3], William Zavadoski[4], Nicole Barucci[4], Marina Godes[1,2], Benjamin M. Moyer[1,2], Callum D. Owen [3], Paul DaSilva-Jardine[5], Donna S. Neuberg [6], Richard A. Bowen [7], Robert A. Davey [3] & Loren D. Walensky [1,2] ✉

The continued emergence of highly pathogenic viruses, which either thwart immune- and small molecule-based therapies or lack interventions entirely, mandates alternative approaches, particularly for prompt and facile pre- and post-exposure prophylaxis. Many highly pathogenic viruses, including coronaviruses, employ the six-helix bundle heptad repeat membrane fusion mechanism to achieve infection. Although heptad-repeat-2 decoys can inhibit viral entry by blocking six-helix bundle assembly, the biophysical and pharmacologic liabilities of peptides have hindered their clinical development. Here, we develop a chemically stapled lipopeptide inhibitor of SARS-CoV-2 as proof-of-concept for the platform. We show that our lead compound blocks infection by a spectrum of SARS-CoV-2 variants, exhibits mucosal persistence upon nasal administration, demonstrates enhanced stability compared to prior analogs, and mitigates infection in hamsters. We further demonstrate that our stapled lipopeptide platform yields nanomolar inhibitors of respiratory syncytial, Ebola, and Nipah viruses by targeting heptad-repeat-1 domains, which exhibit strikingly low mutation rates, enabling on-demand therapeutic intervention to combat viral outbreaks.

The SARS-CoV-2 pandemic has claimed nearly 7 million lives worldwide and over 1.1 million lives in the United States[1], with an unprecedented toll on global populations. The rapid development and deployment of mRNA vaccines, therapeutic antibodies, and diagnostic testing, in addition to repurposing small molecule antivirals and optimizing supportive care, have had a remarkable impact on the course of the pandemic and our ongoing emergence from it. Nevertheless, viral mutagenesis has ensured the persistence of SARS-CoV-2, challenging the efficacy and durability of immunization, rendering early-generation therapeutic antibodies ineffective, and decreasing the impact of small molecule therapeutics, which have essentially been reserved for already ill patients with substantial risk factors. A glaring gap in the therapeutic arsenal for highly pathogenic viruses of pandemic potential, like SARS-CoV-2, is a rapid-response agent that can be easily and immediately administered to prevent and/or ameliorate infection in the form of pre- and post-exposure prophylaxis or to promptly mitigate symptoms upon early infection.

[1]Department of Pediatric Oncology, Dana-Farber Cancer Institute, Boston, MA 02215, USA. [2]Linde Program in Cancer Chemical Biology, Dana-Farber Cancer Institute, Boston, MA 02215, USA. [3]Department of Microbiology, National Emerging Infectious Diseases Laboratories, Boston University, Boston, MA 02118, USA. [4]ATP R&D Labs, Branford, CT 06405, USA. [5]Red Queen Therapeutics, Inc., Cambridge, MA 02142, USA. [6]Department of Biostatistics and Computational Biology, Dana-Farber Cancer Institute, Boston, MA 02215, USA. [7]Department of Biomedical Sciences, Colorado State University, Fort Collins, CO 80523, USA. ✉ e-mail: Loren_Walensky@dfci.harvard.edu

The molecular process that culminates in the fusion of the viral and host membranes is a strikingly conserved mechanism of infection across a host of highly pathogenic viruses. For example, the surface spike glycoprotein of SARS-CoV-2 consists of two subunits, S1 and S2. Upon S1 binding to the host cell ACE2 receptor and exposure to acidified endosomes, conformational changes in the S2 subunit result in the formation of a six-helix bundle (6-HB) that engages and then fuses the host and viral membranes. After fusion, the viral capsid gains access to the host cell cytoplasm, and virus replication ensues. The heptad-repeat 1 and 2 (HR1, HR2) domain α-helices of the SARS-CoV-2 spike protein S2 subunit that form the 6-HB are homologous to those domains found in other viruses such as HIV[2,3], Respiratory Syncytial Virus (RSV)[4], influenza[5,6], Ebola[7,8], Marburg[9], Hendra[10], Nipah[10], and SARS-CoV[11]. A peptide decoy modeled after the HIV-1 gp41 HR2 domain, enfuvirtide (T20), was designed to target HR1 and disrupt the assembly of the 6-HB, thereby blocking HIV-1 entry and infection[12,13]. Enfuvirtide was approved by the FDA as a first-in-class, subcutaneously administered, anti-HIV-1 fusion inhibitor in 2003, but due to its high cost, lack of oral bioavailability for treating a systemic blood-borne infection, and poor in vivo stability, was ultimately relegated to a treatment of last resort. In contrast to the extensive therapeutic options for managing HIV-1 infection, many other viruses that rely on the 6-HB fusion mechanism lack any effective treatments (e.g., Hendra, Nipah, Marburg). Although enfuvirtide provided proof-of-concept, its liabilities have significantly dampened enthusiasm for developing HR2 decoys to tackle hemorrhagic and respiratory viruses of pandemic potential[14–17].

To further develop and optimize this therapeutic strategy for broader application and to address unmet antiviral needs, we initially demonstrated that introducing chemical staples, which crosslink and stabilize α-helical turns of the HR2 peptide, could remedy the liabilities of enfuvirtide. Specifically, double-stapled peptides of the HIV-1 HR2 domain were structurally stabilized, resistant to proteolysis in vitro and in vivo, effectively targeted the HIV-1 fusion bundle, inhibited HIV-1 infectivity, and even overcame enfuvirtide-resistant strains (i.e., escape mutants) owing to the structural pre-folding induced by stapling[18]. Select constructs were so stable that they could be administered orally[18]. We further reported the development of double-stapled respiratory syncytial virus (RSV) F peptides that prevented nasal infection and the spread of the virus to the lungs when administered by the intranasal and intratracheal routes, respectively[19]. These advances demonstrated that HR2 peptides could be engineered to overcome previous limitations and further applied to respiratory infections.

A major concern for the treatment of SARS-CoV-2 infection is the rapid evolution of mutations in the spike protein of variants, thought to be driven by interaction with the human immune response and avoidance of antibody targeting. Unlike the changes on the surface of the glycoprotein spike that allow evasion of neutralizing antibodies and give rise to variants of concern, the 6-HB target is normally buried and highly conserved owing to its critical role in viral entry. The SARS-CoV-2 HR2 domain that forms the basis for the design of peptide decoys shares 100% amino acid identity with the HR2s of other coronaviruses, including SARS-CoV, Bat SARS-like coronaviruses WIV1 and RsSHC014, Bat coronavirus RaTG13, and Pangolin coronavirus. This relatedness suggests that an HR2-based peptide should be broadly active at inhibiting a range of known as well as newly emerging coronaviruses. Whereas multiple reports have described HR2 peptides to inhibit SARS-CoV-2[16,20–26], the limitations exposed by enfuvirtide, particularly with respect to the combination of peptide potency, stability, pharmacology, and clinical viability, have remained unaddressed.

Here, we sought to generate a topical HR2-based agent that inhibits current and future variants of concern to confer on-demand protection against COVID-19. To address prior limitations of candidate HR2 peptides, we develop and evaluate a platform whereby identifying the best site(s) for peptide stapling and appending a lipid moiety with optimal spacing from the stapled peptide yields highly potent, stable, and pharmacologically viable lead compounds for pre- and post-exposure intervention against SARS-CoV-2 and other viruses of pandemic potential.

## Results

### Identification of a lead stapled lipopeptide pan-inhibitor of SARS-CoV-2 variants

To arrest the 6-HB fusion mechanism of SARS-CoV-2, we designed a library of stapled lipopeptides based on the HR2 sequence of SARS-CoV-2 (Fig. 1a, b). Because lipidation has been shown to substantially increase the antiviral activity of HR2 peptide inhibitors of viral fusion[27], we began by developing a facile on-resin synthesis for stably derivatizing HR2 peptides with PEG$n$-thiocholesterol and then used this scaffold as the basis for further refinement (Supplementary Fig. 1a, b). To assess the putative benefit of lipidation, we compared the antiviral activity of a synthetic HR2 peptide (amino acids 1168-1205) with that of a PEG$_4$-thiocholesterol derivative in an in vitro infectivity assay using a SARS-CoV-2 (Wuhan-Hu-1 strain) coronavirus spike protein pseudotyped virus (pseudovirus) and ACE2-expressing HEK293T cells. Whereas the unmodified peptide had no activity within a 40 nM–2.5 μM dosing range, the lipidated construct demonstrated an IC$_{50}$ of 250 nM and IC$_{90}$ of 785 nM (Supplementary Fig. 2).

Using HR2-PEG$_4$-thiocholesterol as a template, we performed an $i, i + 7$ staple scan, whereby an all-hydrocarbon staple spanning two turns of an α-helix is installed sequentially along the length of the peptide sequence to screen for the stapling position that yields optimal bioactivity (Fig. 1c). We tested the 15-member library in infectivity assays using a series of pseudoviruses, including the initial Wuhan-Hu-1 strain, two Omicron variants (B.1.1.529.1 and B.1.1.529.2), and SARS-CoV (Urbani). One of the $i, i + 7$ stapled constructs in particular (D staple) demonstrated high antiviral potency, matching that of the unstapled control peptide at the 250 nM or 500 nM screening dose in the Wuhan-Hu-1 and Omicron pseudoviral infectivity assays, and outperforming it in the SARS-CoV (Urbani) assay (Fig. 1d–g). The D-stapled peptide again emerged as a top hit against SARS-CoV-2 Beta strain (B.1.351) live virus, completely blocking infection of A549-ACE2 cells at the 4 μM screening dose (Fig. 1h), along with several other stapled constructs that exhibited relatively higher potencies in B.1.1.529.1/2 pseudoviral assays as well (B, G, K, L, N) (Fig. 1e, f, h). We then performed a full dose-titration of the consistently best-performing hit, designated Stabilized Alpha-Helix of SARS-CoV-2 HR2 bearing the D staple and PEG$_4$-thiocholesterol (SAH-HR2-D-PEG$_4$-TC), in an expanded panel of pseudoviral assays, which included spikes from Wuhan-Hu-1, B.1 D614G, B.1.1.7 (Alpha), B.1.351 (Beta), B.1.617.2 (Delta), B.1.1.529.1 (Omicron), SARS-CoV (Urbani), and, as a specificity-of-action control, the glycoprotein of vesicular stomatitis virus (VSV-G). SAH-HR2-D-PEG$_4$-TC dose-responsively inhibited each of the SARS-CoV-2 pseudoviral strains, with an IC$_{50}$ range of 42-420 nM (Fig. 1i). Of note, SAH-HR2-D-PEG$_4$-TC had the greatest potency against the Omicron variant, little to no activity against the VSV-G negative control, and up to 1.7-fold improved potency compared to the corresponding unstapled control peptide, which further lacked sigmoidal dose-responses (Supplementary Fig. 3).

Next, we explored the influence of PEG linker length on stapled lipopeptide antiviral activity. We generated a library of SAH-HR2-D-PEG$n$-TC peptides with PEG polymers of up to 20 units. Screening serial dilutions of each construct in infectivity assays using B.1.1.529.1 (Omicron) and SARS-CoV (Urbani) pseudoviruses and Beta strain (B.1.351) live-virus revealed a consistent structure-activity relationship. The shortest PEG lengths of 0 and 3 exhibited the lowest activity, the intermediate PEG lengths of 4–7 had mid-range activity, and PEG lengths of 8 and greater demonstrated the highest antiviral activity (Fig. 2a–c), yielding an order of magnitude improvement in potency over the initial SAH-HR2-D-PEG$_4$-TC hit. These data suggest that the

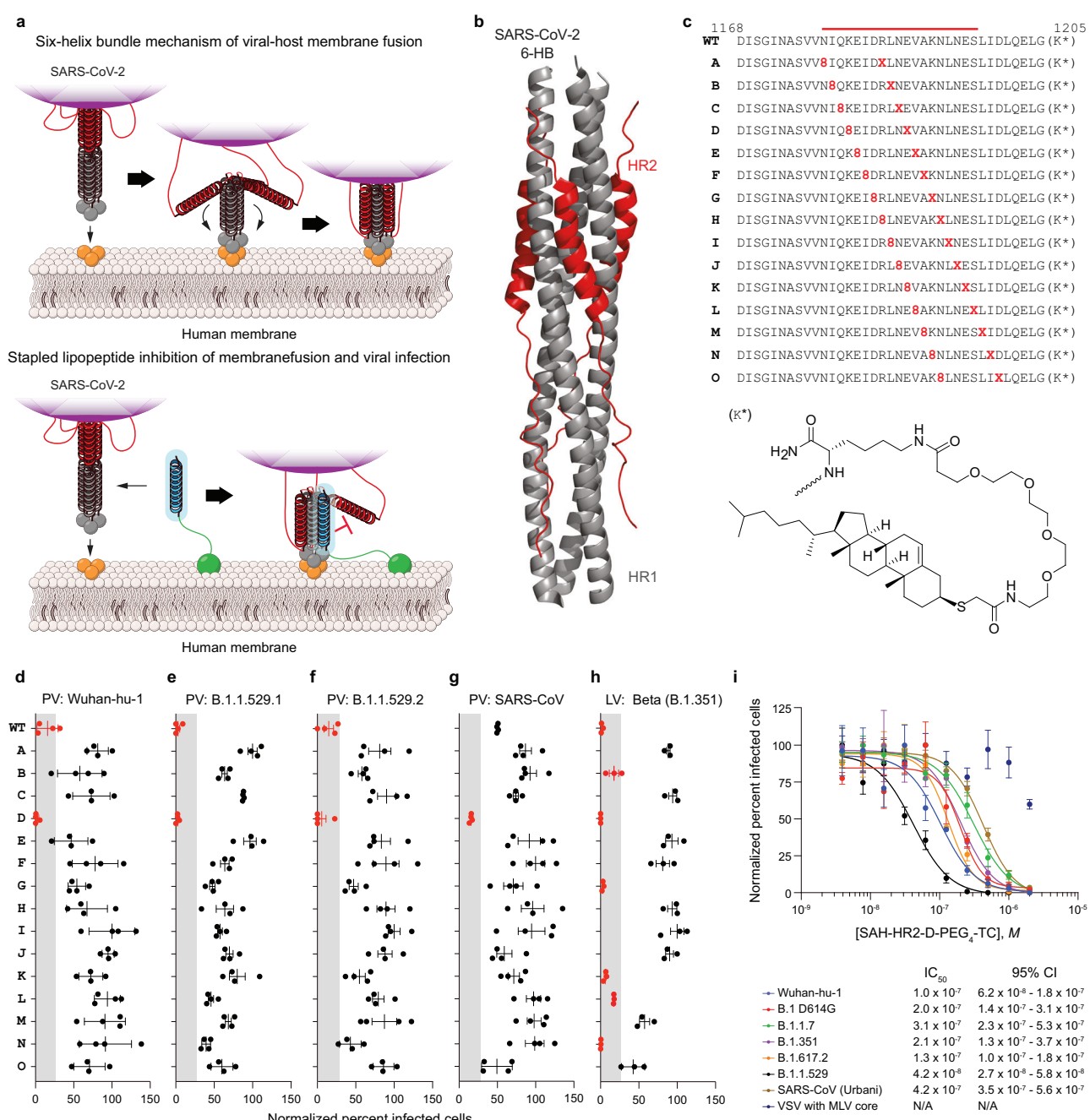

**Fig. 1 | Design and screening of stapled lipopeptide inhibitors of SARS-CoV-2.**
**a** Schematic of the six-helix bundle (6-HB) fusion mechanism of SARS-CoV-2 and the mechanism by which a stapled lipopeptide decoy of the HR2 domain disrupts 6-HB assembly and thus blocks viral entry. **b** Structure of the SARS-CoV-2 6-HB assembly (PDB ID 7TIK), with the HR2 domain that formed the basis for stapled lipopeptide designs colored in red. **c** Compositions of the *i, i + 7* staple scanning library of HR2 amino acid sequences 1168-1205 with the structure of the PEG$_4$-thiocholesterol moiety appended to the C-terminal lysine. WT, unstapled lipo-peptide bearing the indicated wild-type HR2 domain sequence. **d–h** The stapled lipopeptide library was tested in infectivity assays using a series of pseudoviruses (PV), including the initial Wuhan-Hu-1 strain (**d**), Omicron variants B.1.1.529.1 (**e**) and B.1.1.529.2 (**f**), and SARS-CoV Urbani (**g**) in ACE2-expressing HEK293T cells, and SARS-CoV-2 Beta strain live virus (**h**) in ACE2-A549 cells at screening doses of

250 nM (**e**, **f**), 500 nM, (**d**, **g**) or 4 µM (**h**). The data are normalized to the percent infected cells treated with vehicle control. Data are mean ± SEM for assays performed in technical quadruplicate (PV) or triplicate (LV) and then repeated with similar results. The gray shading highlights those lipopeptides that inhibited infectivity to <25% of infected cells (data points colored red). **i** A full dose-titration of the consistently best-performing hit, designated Stabilized Alpha-Helix of SARS-CoV-2 HR2 bearing the D staple and PEG$_4$-thiocholesterol (SAH-HR2-D-PEG$_4$-TC), in an expanded panel of pseudoviral assays, which included Wuhan-Hu-1, D614G, B.1.1.7 (Alpha), B.1.351 (Beta), B.1.617.2 (Delta), B.1.1.529.1 (Omicron), SARS-CoV (Urbani), and VSV-G as a specificity-of-action control. Data are mean ± SEM for assays performed in technical quadruplicate and then repeated with similar results. IC$_{50}$ values were calculated by nonlinear regression analysis of the dose-response curves.

linker has a key influence on performance, presumably by providing optimal flexibility or positioning of the peptide for inhibitory action.

Incorporating the outcomes of iterating staple position and PEG length, and applying our solid phase synthetic schema to replace

thiocholesterol with the less expensive cholesterol moiety (Supplementary Fig. 1 and Supplementary Table 1), we nominated a lead-compound, designated Red Queen Therapeutics 1 (RQ-01), which eluted as a monodispersed peak by fast protein liquid

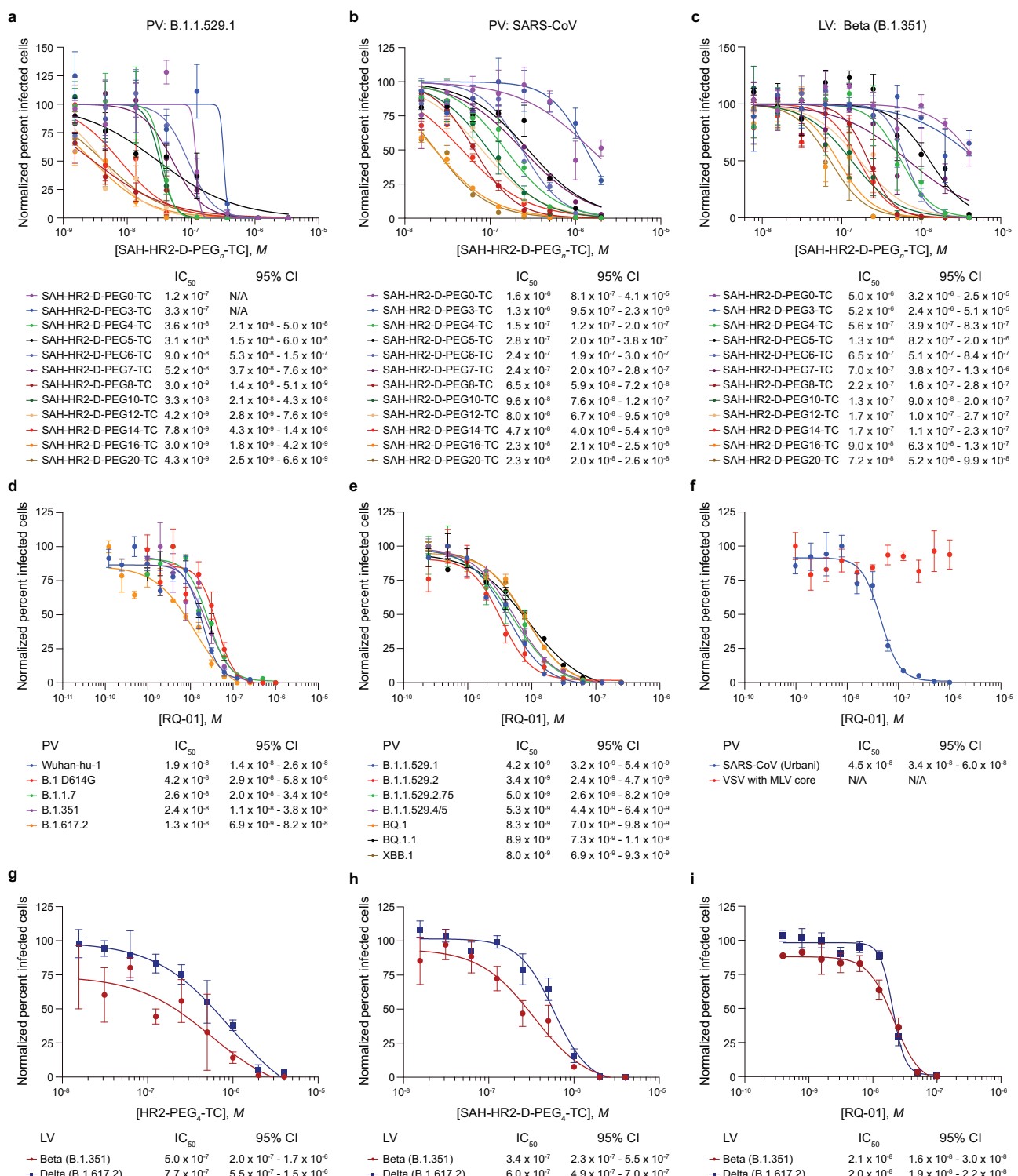

**Fig. 2 | Iterative optimization yields a lead stapled lipopeptide inhibitor of SARS-CoV-2 variants. a–c** A library of SAH-HR2-D-PEGn-TC compounds was generated to assess the impact of PEG linker length (0, 3–8, 10, 12, 14, 16, 20) on antiviral activity. The peptides were tested in infectivity assays using Omicron variant B.1.1.529.1 pseudovirus (**a**), SARS-CoV Urbani pseudovirus (**b**), and Beta strain live virus (**c**). Data are mean ± SEM for assays performed in technical quadruplicate (PV) or triplicate (LV) and then repeated with similar results. IC$_{50}$ values were calculated by nonlinear regression analysis of the dose-response curves. **d–f** Incorporating the outcomes from staple scanning and iterating linker length, a lead compound, designated Red Queen Therapeutics 1 (RQ-01), was developed and subjected to expanded testing in pseudoviral infectivity assays, including Wuhan-

Hu-1, D614G, B.1.1.7 (Alpha), B.1.351 (Beta), and B.1.617.2 (Delta) (**d**); the Omicron variants and sublineages B.1.1.529.1, B.1.1.529.2, B.1.1.529.2.75, B.1.1.529.4/5, BQ.1, BQ.1.1, and XBB.1 (**e**); and SARS-CoV Urbani pseudovirus and the negative control VSV-G virus (**f**). Data are mean ± SEM for assays performed in technical quadruplicate and then repeated with similar results. IC$_{50}$ values were calculated by nonlinear regression analysis of the dose-response curves. **g–i** The relative impact of iterative optimization, as reflected by unstapled HR2-PEG$_4$-TC (**g**), SAH-HR2-D-PEG$_4$-TC (**h**), and RQ-01 (**i**), was then tested in live-virus assays using the SARS-CoV-2 Beta (B.1.351) and Delta (B.1.617.2) strains. Data are mean ± SEM for assays performed in technical triplicate and then repeated with similar results. IC$_{50}$ values were calculated by nonlinear regression analysis of the dose-response curves.

chromatography, exhibited α-helical character by circular dichroism analysis, and localized to the plasma membrane and endosomes of treated cells (Supplementary Fig. 4a–c). We subjected RQ-01 to expanded testing in pseudoviral infectivity assays, including the variants employed in our frontline staple scanning analysis and additional Omicron variants and sublineages such as B.1.1.529.2.75, B.1.1.529.4/5, BQ.1, BQ.1.1, and XBB.1. Interestingly, RQ-01 was more effective against all Omicron variants ($IC_{50}$, 3.4–8.9 nM) than SARS-CoV-2 variants from earlier in the pandemic ($IC_{50}$, 13–42 nM) and SARS-CoV ($IC_{50}$, 45 nM) (Fig. 2d–f), highlighting that mutations which enhanced transmission and immune evasion had no adverse effect on the efficacy of the stapled lipopeptide fusion inhibition mechanism and - if anything - rendered the Omicron variants more susceptible by up to an order of magnitude. Finally, we compared the relative potencies of our starting point, the unstapled HR2-PEG$_4$-TC peptide, and the iteratively optimized stapled constructs, SAH-HR2-D-PEG$_4$-TC and RQ-01 in infectivity assays using live Beta (B.1.351) and Delta (B.1.617.2) SARS-CoV-2 strains. Whereas introducing the staple improved antiviral activity by 1.3 to 1.5-fold compared to the unstapled peptide, the combination of stapling and lengthening the PEG linker (PEG$_8$) resulted in an overall 24 to 39-fold improvement (Fig. 2g–i). Importantly, RQ-01 treatment caused no cytotoxicity and instead enabled the resumption of cell growth upon viral inhibition (Supplementary Fig. 5a, b). Thus, our facile on-resin production of stapled lipopeptide libraries enabled the discovery of a lead construct with low nanomolar pan-inhibitory activity against SARS-CoV-2 variants in pseudo- and live-virus assays.

## RQ-01 structure-activity relationship and comparative enhancement of stability and antiviral potency

To further evaluate the relative contributions of key features of a lead stapled lipopeptide inhibitor of SARS-CoV-2, we performed a series of characterization assays. First, we measured by AlphaScreen the affinity of RQ-01 for its binding target, the 5-helix bundle (5-HB) of SARS-CoV-2 comprised of 3 HR1 and 2 HR2 domains. Notably, the binding affinity of RQ-01 and the corresponding analogs lacking either the lipid moiety (RQ-01 [-lipid]) or both the PEG linker and lipid moieties (RQ-01 [-linker-lipid]) exhibited essentially the same binding activities ($EC_{50}$, 0.96–1.9 nM) (Fig. 3a). These results indicate that the ligand for 5-HB interaction, the stapled HR2 sequence, is the only contributor to the observed binding activities, with incorporation of the staple improving affinity by 7.5-fold compared to the corresponding unstapled HR2 peptide (Supplementary Fig. 6). In contrast, when the same trio of peptides was tested in the Omicron BA.1.1.529.2 pseudoviral infectivity assay, only RQ-01 exhibited potent antiviral activity ($IC_{50}$, 3 nM), outperforming the corresponding constructs lacking lipid or the linker-lipid by 3 orders of magnitude (Fig. 3b). These data indicate that, in the cellular context, lipidation is driving a substantial enhancement of antiviral activity, presumably by concentrating the peptide at the membrane surface—the mechanistic site of action.

To further examine the relevance of stapling, we compared RQ-01 to a previously reported lead compound, SARS$_{HRC}$-PEG$_4$-chol (where the heptad-repeat C-terminus [HRC] sequence corresponds to aa 1168-1203 of the SARS-CoV-2 HR2 domain), which exhibited essentially identical potency in fusion inhibition assays as the next-generation compounds SARS$_{HRC}$-PEG$_{24}$-chol (extended PEG linker) and [SARS$_{HRC}$-PEG$_4$]$_2$ (a dimer analog of SARS$_{HRC}$-PEG$_4$-chol)[20]. In infectivity assays using Delta (B.1.617.2) and Omicron (B.1.1.529.1) pseudoviruses and the live Delta (B.1.617.2) virus, RQ-01 consistently outperformed SARS$_{HRC}$-PEG$_4$-chol by 4.7-, 23-, and 5.4-fold respectively, with the greatest differential observed against the Omicron variant (Fig. 3c–e). Given that the main compositional difference between RQ-01 and SARS$_{HRC}$-PEG$_4$-chol is the incorporation of the all-hydrocarbon $i, i+7$ staple (Supplementary Table 1), we examined the relative stability of the two compounds upon exposure to acid, base, and heat. Monitoring intact compounds by liquid chromatography–mass spectrometry (LC/MS),

we observed the decomposition of SARS$_{HRC}$-PEG$_4$-chol to 6-30% of initial levels, whereas RQ-01 was fully preserved (Fig. 3f–h). Finally, another key requirement for the translation of lipopeptides is solubility, with marked insolubility in aqueous solution noted for previously reported unstapled lipopeptides[20]. In comparing the solubility of RQ-01 and SARS$_{HRC}$-PEG$_4$-chol in 50 mM sodium phosphate buffer, pH 7.0 across a broad 15-90 mg/mL concentration range, we observed complete solubility of RQ-01 and no solubility of SARS$_{HRC}$-PEG$_4$-chol (Supplementary Fig. 7). Taken together, these data indicate that (1) the peptide sequence of RQ-01 drives target binding affinity, (2) lipidation markedly enhances antiviral potency in the cellular context, and (3) stapling contributes in part to target binding affinity and antiviral potency and even more strikingly to compound stability and solubility. Indeed, it is the unique combination of these compositional features that yields an optimized stapled lipopeptide inhibitor of SARS-CoV-2 for in vivo application.

## In vivo stability and antiviral efficacy of RQ-01

In advance of in vivo efficacy testing, we undertook a tissue pharmacokinetic (PK) study in which male mice aged 6-7 weeks were anesthetized by inhaled isoflurane and treated with 1, 3, or 10 mg/kg of RQ-01 by nasal drop, as formulated in a mucoadhesive solution (0.1% hydroxypropyl methylcellulose [HPMC]/50 mM sodium phosphate, pH 6.5). After compound administration, mice were sacrificed at timed intervals followed by a blood draw and excision of nasal and lung tissues for prompt processing. Peptide levels were detected and quantitated by LC/MS-MS. After a single intranasal dose, nasal tissue levels could be sustained at over 100x the infectivity assay IC90 values for at least 24 h (Fig. 4a). Peak lung tissue concentrations were approximately 1/10th of those measured in nasal tissue, whereas plasma exposure was minimal at approximately 1/1000th of that measured in nasal tissue, consistent with our goal of localized topical delivery of RQ-01 at the initial site of SARS-CoV-2 infection[28] (Supplementary Fig. 8a). A PK study comparing nasal and lung tissue levels out to 48 h after a 2 mg/kg intranasal dose formulated in a saline buffer (0.9% saline, pH 7.2) again demonstrated sustained nasal tissue exposure, decreasing from 10 μM to 1 μM by 48 h, whereas lung levels peaked between 2 and 4 h at 1 μM (Fig. 4b). Prior to finalizing our dose selection for in vivo testing in a hamster model of SARS-CoV-2 infection, we confirmed that an intranasally administered dose could achieve similar nasal and lung tissue concentrations in hamsters and mice out to 24 h (Supplementary Fig. 8b). Based on these tissue PK data, we selected a 3 mg/kg dose in the mucoadhesive formulation for in vivo efficacy testing.

To assess the impact of RQ-01 treatment on the prevention and treatment of SARS-CoV-2 in a small animal model, Syrian hamsters (140 g average weight) were challenged with the WA-1 (Wuhan) isolate of SARS-CoV-2 under ketamine-xylazine anesthesia by intranasal instillation of 100 μL of $9.8 \times 10^3$ plaque-forming units (PFU) of the virus. Hamsters ($n = 8$ per group; $n = 4$ per sex) were either mock-infected (Group 1), treated with vehicle (Group 2) or 3 mg/kg RQ-01 (100 μL/nare) in accordance with the following dosing schedules: −0.5, 12, 24, 48, 72 h (Group 3); −0.5, 24, 48, 72 h (Group 4); 12, 24, 48, 72 h (Group 5); 24, 48, 72 h (Group 6). Outcome measures included daily weight, oropharyngeal (OP) swab titers (Days 1–4), and histopathologic evaluation of the lungs (Day 4). A significant protective effect from weight loss was observed for peptide treatment Groups 3-5, each at $p < 0.0001$; protection from weight loss was also observed in Group 6 but with $p = 0.06$ (Fig. 5a, b). Whereas hamsters that received pre-treatment and early post-treatment (Groups 3–5) exhibited relatively stable weights from the outset and then resumed weight gain at Day 2, hamsters that did not receive their first treatment dose until 24 h after viral inoculation initially lost weight, in accordance with vehicle-treated mice, but then weight loss ceased within 24 h of receiving RQ-01. Viral

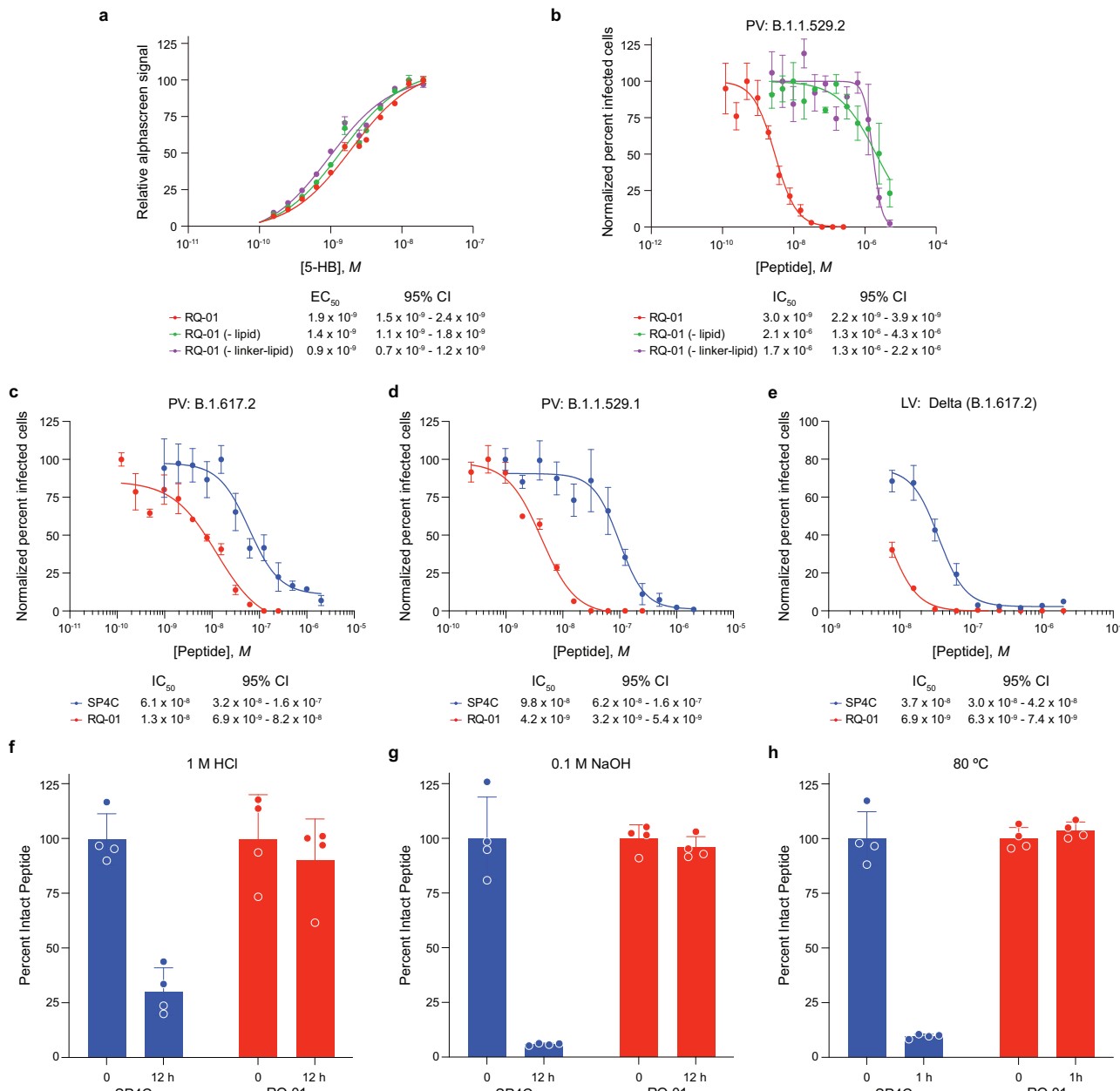

**Fig. 3 | Stapling and lipidation confer superior biophysical and antiviral properties independently and in combination. a** AlphaScreen assay demonstrating the binding activity of RQ-01, and its analogs lacking either the lipid or the linker-lipid combination, for its target, the 5-helix bundle (5-HB) of SARS-CoV-2 comprised of 3 HR1 and 2 HR2 domains. Data are mean ± SEM for assays performed in technical quadruplicate and then repeated with similar results. IC$_{50}$ values were calculated by nonlinear regression analysis of the dose-response curves. **b** The comparative antiviral activity of the same trio of compounds was tested in infectivity assays using Omicron variant B.1.1.529.2 pseudovirus in ACE2-expressing HEK293T cells. Data are mean ± SEM for assays performed in technical quadruplicate and then repeated with similar results. IC$_{50}$ values were calculated by nonlinear regression analysis of the dose-response curves. **c**–**e** RQ-01 (lipidated/stapled) and SARS$_{HRC}$-PEG$_4$-chol (SP4C) (lipidated/not stapled) were evaluated in infectivity assays using pseudoviruses B.1.617.2 (**c**) and B.1.1.529.1 (**d**) and the live Delta (B.1.617.2) virus (**e**). Data are mean ± SEM for assays performed in technical quadruplicate (PV) or triplicate (LV) and then repeated with similar results. IC$_{50}$ values were calculated by nonlinear regression analysis of the dose-response curves. **f**–**h** The comparative stability of RQ-01 and SARS$_{HRC}$-PEG$_4$-chol (SP4C) upon exposure to acid, base, and extreme heat (80 °C) was assessed by HPLC-based detection and quantitation. Data are mean ± SEM for assays performed in technical quadruplicate and then repeated with similar results.

shedding as monitored by OP swab was significantly reduced by the regimens that included pre-treatment (Groups 3, 4), as evidenced by both Day 1 (Fig. 5c) and the sum of swabs (Days 1–4) (Fig. 5d) data. The histopathologic evaluation further revealed significant protection from the pulmonary sequelae of SARS-CoV-2 infection, as demonstrated by the reduction in pneumonia global severity score (Fig. 5e) and the total lung score (Fig. 5f), which comprises assessments of pneumonia global severity, bronchointerstitial

pneumonia, type II pneumocyte hyperplasia, bronchiolar degeneration/necrosis, bronchiolar hyperplasia, and vasculitis/endotheliitis.

The notable preservation of hamster health, as reflected by protection from weight loss, was further assessed in a SARS-CoV-2 transmission model. An infected donor cohort (*n* = 12) was inoculated intranasally on Day 0 with 100 μL of 1.1 x 10$^4$ PFU of WA-1 SARS-CoV-2 isolate (Group 1) and then each donor was placed in a cage with two

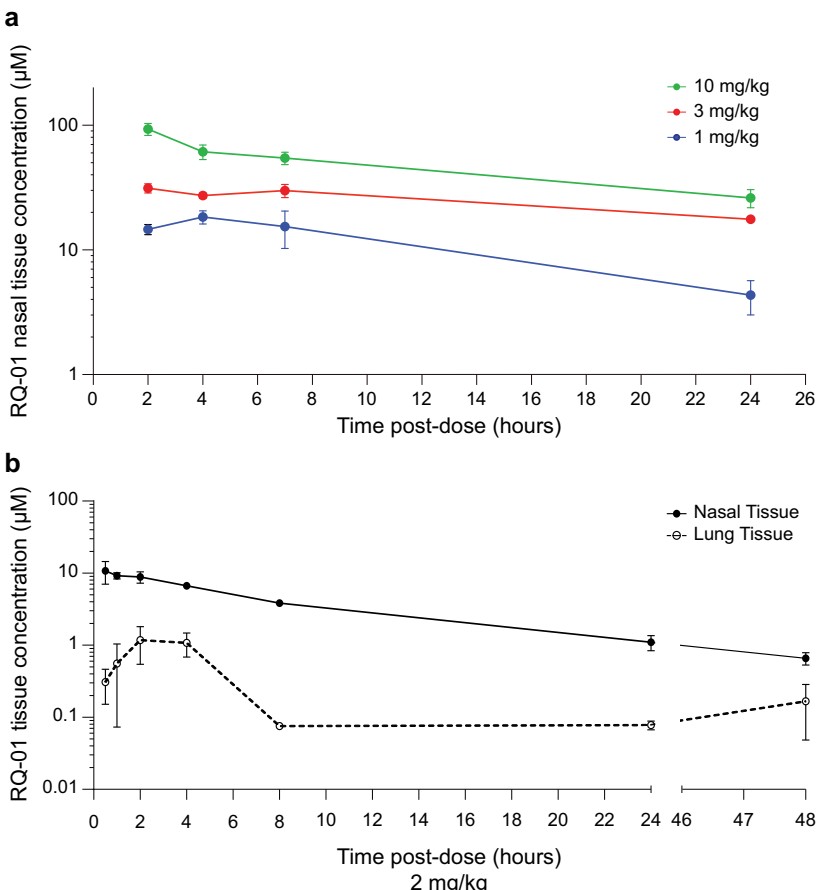

**Fig. 4 | Tissue persistence of RQ-01 administered intranasally. a** Mice were treated intranasally with 1, 3, or 10 mg/kg of RQ-01 formulated in 0.1% HPMC/50 mM sodium phosphate, pH 6.5 and sacrificed at 2, 4, 7, or 24 h for PK analysis, performed by LC/MS-MS of nasal tissue homogenates. Data are mean ± SEM for drug levels quantified for $n = 3$ mice per dosing level and time point. **b** Mice were treated intranasally with 2 mg/kg of RQ-01 formulated in 0.9% saline, pH 7.2, and sacrificed at 0.5, 1, 2, 4, 8, 24, and 48 h after dosing for PK analysis, performed by LC/MS-MS of nasal and lung tissue homogenates. Data are mean ± SEM for drug levels quantified for $n = 3$ mice per dosing level and time point.

sentinel hamsters on Day 1 for a 24-h exposure period, after which the donor was removed and the two sentinels separated and housed in individual cages. Male sentinel hamsters ($n = 8$ per group) were treated with either vehicle (Group 2) or 3 mg/kg RQ-01 (100 μL/nare) in accordance with the following dosing schedules, which are relative to initial donor exposure on Day 1: −2, 24, 48, 72 h (Group 3); 24, 48, 72 h (Group 4). Whereas the donor and vehicle-treated sentinel cohorts sustained daily progressive weight loss with similar slopes, both groups of RQ-01 treated sentinels—whether pre-treated or not—demonstrated protection from weight loss, each at $p < 0.0001$ (Supplementary Fig. 9a, b). A subset of hamsters from both RQ-01 treated cohorts exhibited reduced OP swab titers (Day 3) and both RQ-01 treated cohorts demonstrated suppression of lung viral titers compared to sentinels that received vehicle (Supplementary Fig. 9c, d). Thus, whereas RQ-01 treatment with 3 mg/kg dosing did not prevent transmission to sentinel hamsters, the sequelae of viral exposure were substantially reduced compared to vehicle-treated sentinels. Taken together, the in vivo data demonstrate that RQ-01 can achieve tissue persistence upon intranasal administration to mice and can mitigate the clinical consequences of SARS-CoV-2 infection in hamsters whether administered before or after (1) direct viral inoculation or (2) exposure to an infected donor. Indeed, the results highlight the potential of RQ-01 to serve as an effective agent for both prophylaxis against and reducing symptoms of SARS-CoV-2 infection.

## The stapled lipopeptide platform yields potent inhibitors of respiratory syncytial, Ebola, and Nipah viruses

To demonstrate the broader utility of our stapled lipopeptide platform in rapidly developing lead inhibitors for pathogenic viruses, we applied our workflow to the HR2 sequences of RSV, Ebola, and Nipah viruses (Fig. 6a–c). In each case, staple scanning identified optimal staple position(s) for inhibiting RSV and Ebola live viruses and Nipah pseudovirus in in vitro infectivity assays (Fig. 6d–f). Then, additional iteration, including sampling differential PEG-lipid linker lengths, produced lead stapled lipopeptides with low nanomolar antiviral activity and no observed cytotoxicity for development as novel agents to prevent and treat RSV, Ebola, and Nipah infections (Fig. 6g–i and Supplementary Fig. 5c–e). In this manner, stapled lipopeptide fusion inhibitors can be optimized to arrive at a lead compound within weeks of HR2 sequence identification. As demonstrated here for RQ-01, PK studies and in vivo efficacy testing in animal models can then provide proof-of-concept for clinical development (Fig. 6j). Ultimately, the unique features of stapled lipopeptides, including thermal stability for facile transport and storage and the capacity for on-demand treatment—whether delivered intranasally or potentially by nebulization for respiratory viruses or subcutaneously for hemorrhagic fever viruses—could help combat outbreaks driven by emerging and established pathogens and their variants, particularly when no other treatments exist and urgent prophylaxis and treatment are required.

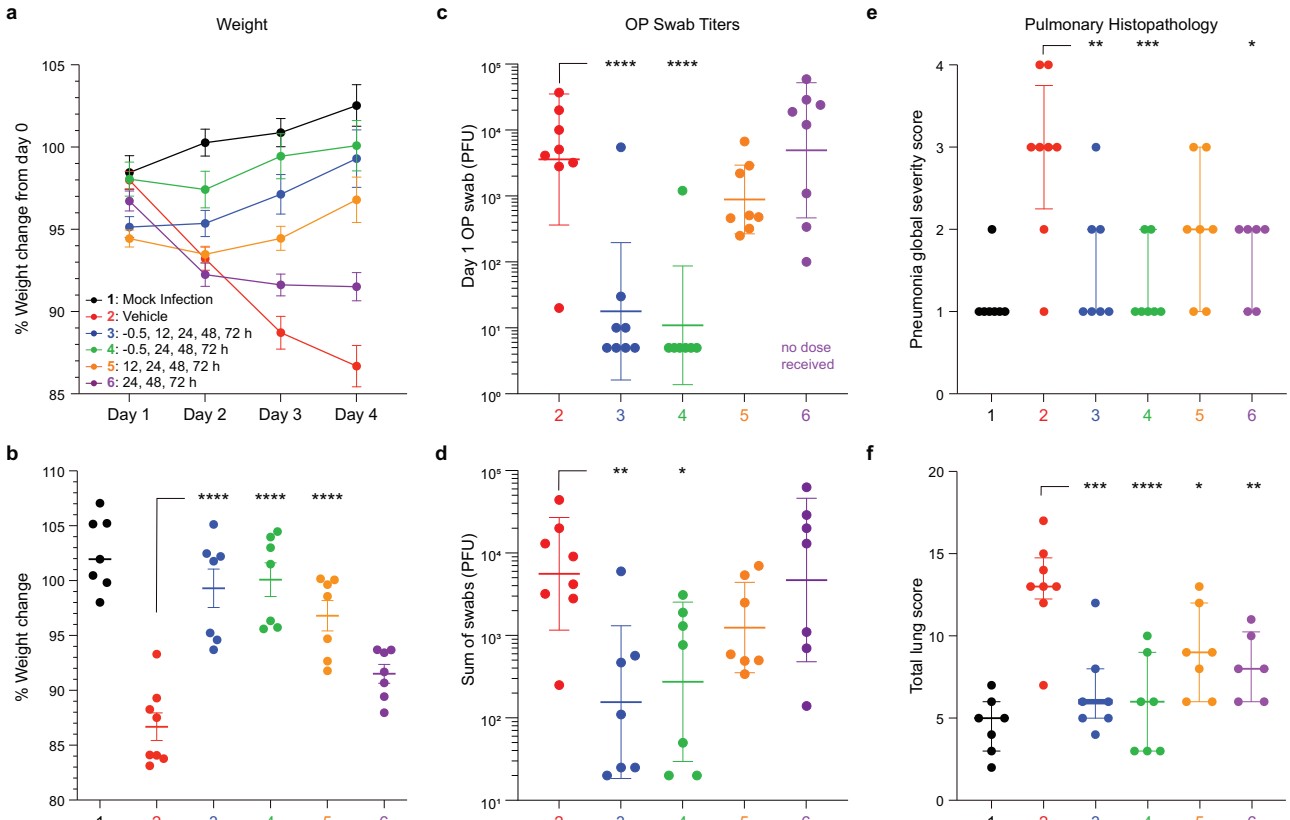

**Fig. 5 | In vivo efficacy of RQ-01 in a hamster model of SARS-CoV-2 infection.**
**a**–**f** Syrian hamsters were either mock-infected (Group 1) or challenged by intranasal instillation of 100 μL of the WA-01 (Wuhan) isolate containing $9.8 \times 10^3$ plaque-forming units (PFU) of the virus at time 0 h (Groups 2–6). Hamsters were either treated intranasally with vehicle (Group 2) or 3 mg/kg RQ-01 in accordance with the following dosing schedules: −0.5, 12, 24, 48, 72 h (Group 3); −0.5, 24, 48, 72 h (Groups 1 and 4); 12, 24, 48, 72 h (Group 5); 24, 48, 72 h (Group 6). Plotted comparative outcome measures include daily percent initial (Day 0) weight (Days 1–4) (**a**), percent initial (Day 0) weight on Day 4 (Groups 3-5, $p < 0.0001$; Group 6, $p = 0.060$) (**b**), oropharyngeal (OP) swab titer from Day 1 (Groups 3–4, $p < 0.0001$;

Group 5, $p = 0.49$; Group 6, $p = 0.99$) (**c**), the sum of swab titers from Days 1–4 (Group 3, $p = 0.004$; Group 4, $p = 0.017$; Group 5, $p = 0.38$; Group 6, $p = 0.99$) (**d**), and pneumonia global severity score (Group 3, $p = 0.005$; Group 4, $p = 0.0006$; Group 5, $p = 0.090$; Group 6, $p = 0.014$) (**e**) and total lung score (Group 3, $p = 0.0001$; Group 4, $p < 0.0001$; Group 5, $p = 0.020$; Group 6, $p = 0.005$) (**f**) based on blinded histopathologic evaluation. For outcomes measured in $n = 8$ mice per treatment arm, data are mean ± SEM (**a**, **b**), geometric mean ± geometric SD (**c**, **d**), and median with interquartile range (**e**, **f**). One-way ANOVA with Dunnett's Multiple Comparisons Test to Vehicle: \*\*\*\*$p < 0.0001$; \*\*\*$p < 0.001$; \*\*$p < 0.01$; \*$p < 0.05$.

## Discussion

The administration of a peptide, enfuvirtide, to block HIV-1 infection by preventing membrane fusion and thus viral entry is a clinically validated mechanism of action dating back twenty years[29]. However, the need for multiple subcutaneous injections per day, injection site reactions, and resultant patient noncompliance, among other challenges (e.g., cost of goods, peptide instability, suboptimal pharmacology, replacement by a myriad of other HIV-1 treatments), rendered enfuvirtide a treatment of last resort for HIV-1/AIDS. In contrast to HIV-1/AIDS, there are a host of respiratory and hemorrhagic fever viruses that have no known treatments yet also depend on the identical six-helix bundle mechanism of viral infection[30]. Unlike vaccines that require a waiting period for efficacy, depend on a healthy immune system capable of mounting a vaccine response, and have a proven risk of waning immunity[31], a topically or subcutaneously administered fusion inhibitor that targets one of the least mutated regions of the viral spike[32] seems worthy of clinical translation. Why then has there been little to no investment in adapting this proven fusion inhibitor modality as a first-in-class prophylactic or treatment for coronaviruses, filoviruses, and paramyxoviruses of pandemic potential?

A significant hurdle for developing peptide therapeutics derives from their long-established vulnerability to degradation in vivo and poor pharmacology[33]. However, new techniques, such as lipidation

and hydrocarbon stapling, have proven not only to extend the stability and half-lives of peptide therapeutics, but also to completely alter the traditional mode of peptide drug clearance[34,35]. For example, lipidation of the GLP-1 peptide sequence, in the form of liraglutide, for the treatment of diabetes transformed the 1–1.5 h in vivo half-life of the native sequence to 11–15 h when administered subcutaneously[36]. The further enhancement of albumin binding affinity and blockade of dipeptidyl peptidase IV (DPP-4) proteolysis by amino acid substitution combined to endow the lipopeptide semaglutide with a half-life of one week, making it viable for clinical use[36]. An alternative approach that involved introducing all-hydrocarbon stitches into the GLP-1 sequence, such that two structurally stabilizing staples are formed via a central *bis*-bridging stapling amino acid, yielded a compound with 23-fold greater ex vivo serum stability than the natural GLP-1 peptide, compared to the 12-fold stability enhancement demonstrated by semaglutide; interestingly, the stitched construct correspondingly produced a more rapid, initial reduction in serum glucose than semaglutide when administered to diabetic Lepr[db] mice[37]. A hydrocarbon-stapled p53 peptide was found to have marked stability in humans and exhibit hepatobiliary excretion, resulting in an extended half-life compared to the otherwise rapid renal clearance typically observed for native peptides[38]. The benefits of installing staples into native sequences to reinforce bioactive alpha-helical structure, optimize peptide stability, and enhance biological activity have also been

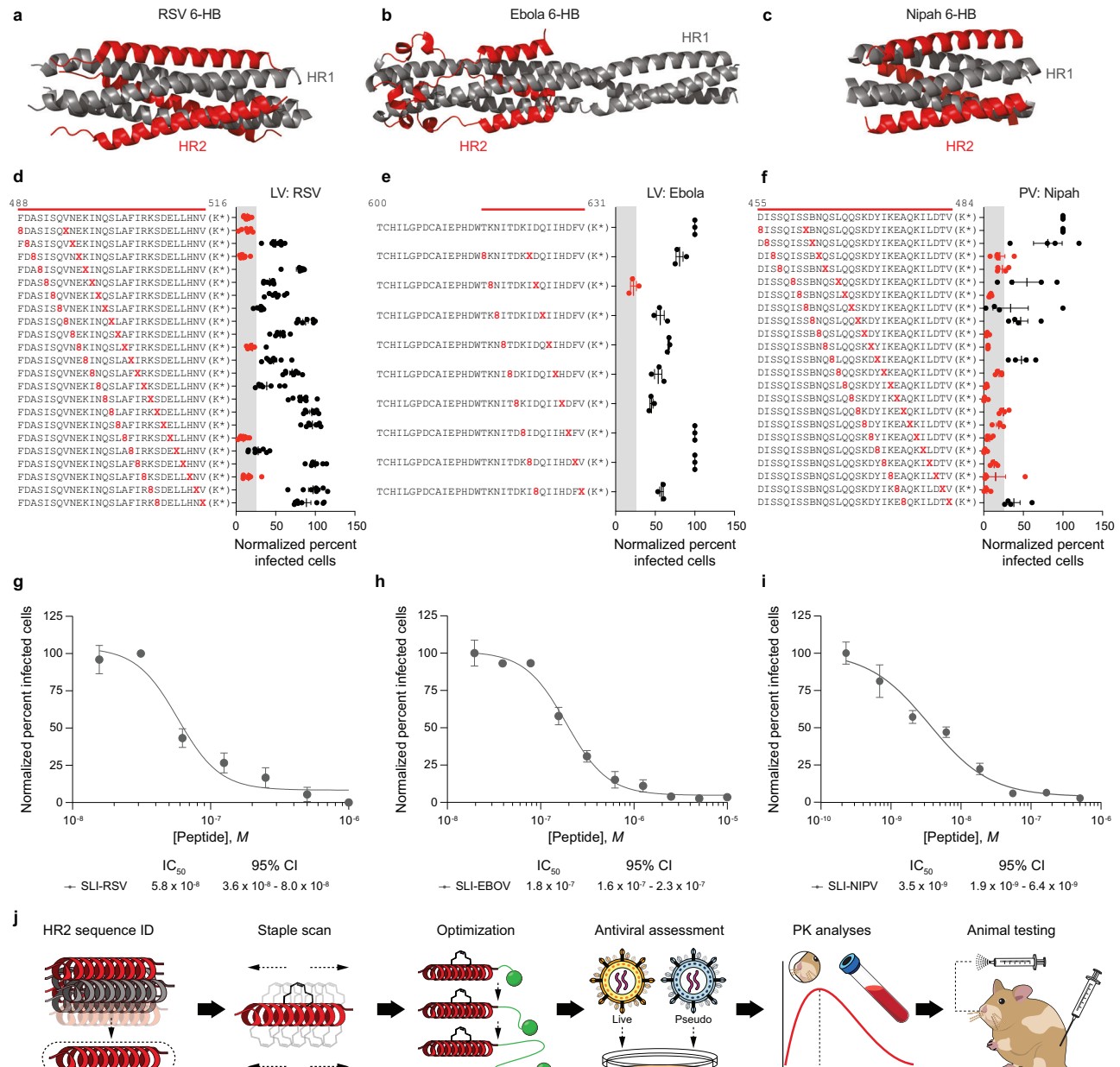

**Fig. 6 | Development of lead stapled lipopeptide inhibitors of RSV, Ebola, and Nipah viruses. a–c** Structures of the conserved 6-HB assemblies of RSV (**a**), Ebola (**b**), and Nipah (**c**) viruses (PDB IDs 1G2C, 1EBO, and 1WP7, respectively), with the HR2 domain that formed the basis for stapled lipopeptide inhibitor (SLI) designs colored in red. **d–f** Staple scanning panels of RSV (**d**), Ebola (**e**), and Nipah (**f**) viruses were tested in the corresponding in vitro infectivity assays against RSV (A549 cells) and Ebola (HeLa cells) live viruses and Nipah (293T cells) pseudovirus at screening doses of 2, 10, or 5 μM, respectively. Data are mean ± SEM for assays performed in at least technical triplicate (RSV, n = 8; Ebola, n = 3; Nipah, n = 4), and then repeated with similar results. The gray shading highlights those lipopeptides that inhibited infectivity to <25% of infected cells (data points colored red). **g–i** Further

optimization, including iteration of PEG-lipid linker length, yielded lead SLIs with low nanomolar activity against RSV (A549 cells) and Ebola (HeLa cells) live viruses and Nipah (293T cells) pseudovirus, as assessed by in vitro infectivity assays. Data are mean ± SEM for assays performed in at least technical triplicate (RSV, n = 4; Ebola, n = 3; Nipah, n = 10) and then repeated with similar results. $IC_{50}$ values were calculated by nonlinear regression analysis of the dose-response curves. **j** A workflow for the development of SLIs for highly pathogenic viruses, incorporating staple scanning of HR2 domain sequences, iterative optimization of PEG-lipid composition, biophysical analyses, functional testing in in vitro pseudovirus and live-virus assays, PK analyses in respiratory tissues and blood, and in vivo efficacy testing in animal models of infection and transmission.

demonstrated for the HR2 domains of HIV-1 and RSV. In each case, we found that stapled analogs of the HIV-1 and RSV HR2 domains exhibited half-lives of up to 3 orders of magnitude greater than those of the corresponding unstapled peptides upon protease exposure, with peptide alpha-helicity and antiviral activity also enhanced[18,19]. Not surprisingly, the potential to harness HR2 domain peptides as a therapeutic strategy for COVID-19 led to numerous reports of peptide and lipopeptide designs for blocking the fusion apparatus of SARS-CoV-2[16,20–26] and select constructs were advanced to small animal testing

with encouraging results[16,20]. Nevertheless, clinical translation of such unstapled constructs has not occurred, likely due to the typical roadblocks of lengthy peptide drug development, including the cost of goods, instability, insolubility, suboptimal pharmacology, and the associated lack of pharma interest. Here, we combined lipidation and all-hydrocarbon stapling in an effort to overcome the formidable barriers to clinical translation of HR2 peptide-based antivirals, given the urgent need for agents that block a host of highly pathogenic viruses, whether respiratory or blood-borne.

Here, we find that stapled lipopeptide fusion inhibitors hold promise to address an unmet need in the management of COVID-19: a self-administrable, on-demand intervention that operates across SARS-CoV-2 variants (owing to the high conservation of the 6-HB fusion apparatus) as pre-exposure prophylaxis, post-exposure prophylaxis, and treatment. The combined attributes of stapling and lipidation yielded a structurally stable lead compound, RQ-01, which exhibits low nanomolar potency across all SARS-CoV-2 strains tested to date. Compared to previously reported HR2-based constructs that were tested in small animals[16,20], RQ-01 exhibits markedly improved peptide stability and solubility, enhanced potency and breadth of antiviral activity, and fully on-resin chemical synthesis, including the lipidation step, thereby streamlining synthesis and purification for facile iteration, lead selection, and upscale. The local tissue persistence of nasally administered compound showcases the desirable pharmacologic properties of stapled lipopeptides, with RQ-01 protecting hamsters from SARS-CoV-2-induced weight loss and pulmonary damage, whether administered before or after viral inoculation. The benefits of RQ-01 treatment were likewise evident in a transmission model of SARS-CoV-2 infection. Delivery approaches such as nebulization that enhance lung exposure also warrant evaluation. Thus, we believe that stapled lipopeptides provide an alternative and complementary therapeutic modality for COVID-19, with compelling advantages. Specifically, an effective anti-SARS-CoV-2 nose spray would address the challenges of (1) vaccine compliance; (2) delay in achieving immunity post-vaccination; (3) vaccine and monoclonal antibody immune evasion due to spike protein mutations; (4) protecting immunocompromised patients; and (5) the unmet need for rapid and facile pre- and post-exposure prophylaxis. It is noteworthy that RQ-01 is resistant to acid, base, and extreme heat, attributes particularly amenable to transport and storage, including over the long term, as needed for stockpiling.

Because so many viral families rely on the six-helix bundle fusion mechanism to achieve host infection[2–11], we envision that our stapled lipopeptide platform can be harnessed to advance topical and systemically administered antivirals for a broad spectrum of respiratory and hemorrhagic fever viruses of pandemic potential, as demonstrated by our rapid development of lead stapled lipopeptide inhibitors of RSV, Ebola, and Nipah viruses. Such stapled lipopeptides could be a critical addition to the therapeutic arsenal so that future outbreaks can be combated by compounds-in-hand that can promptly block or mitigate infection and eliminate or reduce viral transmission. Indeed, our approach to combining lipidation and hydrocarbon stapling, particularly for optimizing the stability, solubility, pharmacology, and biological activity of α-helical peptides, could enable the advancement of a new generation of peptide therapeutics for clinical applications in infectious diseases, diabetes, cancer, and a host of other conditions.

## Methods

### Ethical regulations statement
The murine PK study was conducted under approved Institutional Animal Care and Use Committee (IACUC) protocol VAS-103 at ATP Research & Development (Branford, CT). PK analysis and in vivo efficacy testing in hamsters were conducted under Colorado State University's approved IACUC protocol #1035 and Institutional Biosafety Committee protocol #20-029B.

### Stapled lipopeptide synthesis
All-hydrocarbon stapled peptides were synthesized by solid phase Fmoc chemistry on rink-amide resin (Sigma). First, an orthogonally protected lysine, Fmoc-L-Lys(Dde)-OH (Chem-impex) was incorporated at the C-terminus. Elongation of the peptides was then performed with hexafluorophosphate azabenzotriazole tetramethyl uronium (HATU, Oakwood Chemical) as the coupling reagent and N,N-diisopropylethylamine (DIEA) as the base. Pairs of amino acids at discrete $i, i + 7$ positions were replaced with the (R)-N-Fmoc-α-(7-octenyl) alanine and (S)-N-Fmoc-α-(4-pentenyl)alanine non-natural amino acids (Nagase) at the indicated locations. Upon completion of the primary sequence, the N-terminus was acetylated with acetic anhydride, the Dde protecting group was removed with 2% hydrazine in DMF (5 x 2 min) and a linker, Fmoc-PEG$n$-CO$_2$H (ChemPep), was coupled to the liberated amine followed by olefin metathesis using the Grubb's first-generation catalyst (3 x 2 h). Lastly, after Fmoc deprotection, 2-(thiocholesterol)acetic acid or 2-(cholesterol)acetic acid (see synthesis below) was coupled to the PEG linker and the peptide was cleaved from the resin by a solution of 95:2.5:2.5 (v/v) TFA/TIS/water and purified to >95% purity by C18 reverse phase high performance liquid chromatography (HPLC) using an acetonitrile/water gradient supplemented with 0.1% formic acid. N-terminal derivatization with Cy5.5-β-Ala or biotin-PEG$_2$ yielded conjugates for cellular imaging and AlphaScreen, respectively. The purified peptides were lyophilized, quantified by amino acid analysis, and stored at −20 °C. A listing of all peptide compositions generated for this study can be found in Supplementary Table 1.

### Synthesis of 2-(sterol)acetic acid
To functionalize the sterol with a carboxy handle, a simple two-step, one-pot synthesis was developed, whereby the sterol was dissolved in solvent at 0.1 M (DCM for thiocholesterol or THF for cholesterol) in a round bottom flask, 5 eq of base (DIEA for thiocholesterol or KOtBu for cholesterol) was added, followed by 3 eq of the t-butyl ester of bromoacetic acid, and the reaction stirred overnight at room temperature. Two volumes of trifluoroacetic acid (relative to DCM) were then added and the reaction was stirred at room temperature for 30 min. The reaction progress was monitored by thin-layer chromatography (TLC) (19:1 Hex:EtOAc) with KMnO$_4$ staining. Thiocholesterol migrated with the solvent front with the thioether slowing migration by ~20% and TFA hydrolysis bringing the TLC spot to the baseline. The reaction mixture was added to 5 vol water, followed by the addition of 1 vol DCM. The DCM layer was washed sequentially with 0.1 M HCl and brine and then dried with sodium sulfate. Removal of the solvent by Rotovap yielded an orange heavy oil that was used without further purification. The yield was near quantitative. Purity was determined to be greater than 90% by NMR based on the integration of the sterol olefin proton and the new CH$_2$ singlet. All chemical reagents used for the synthesis were purchased from Sigma-Aldrich.

### Pseudovirus assays
To assess antiviral activity against SARS-CoV-2 pseudotyped viruses with GFP reporters, 293T-hACE2 (Integral Molecular Cat# C-HA101) cells were used and the variants tested included Wuhan-Hu-1, B.1 D614G, B.1.1.7, B.1.351, B.1.617.2, B.1.1.529.1, B.1.1.529.2, B.1.1.529.2.75, B.1.1.529.4/5, BQ.1, BQ.1.1, XBB.1, and SARS-CoV Urbani. (Integral Molecular Cat# RVP-701, -702, -706, -724, -763, -768, -770, -776, -774, -785, -782, -780, and -801). For testing activity against the Nipah pseudotyped virus, 293T cells and the Nipah variant, Malaysia 2008, were used (Integral Molecular Cat# RVP-1801). Specificity of action was assessed using a negative control pseudovirus, VSV-G Indiana strain with MLV core (Integral Molecular Cat# RVP-1002). A single dose or serial two-fold dilution of peptide was incubated with 0.25–1.25 μL of pseudotyped GFP virus in 5 μL of water for 20 min at 37 °C in a 384-well black clear-bottom plate, followed by addition of 35 μL of 1000 cells (293T-hACE2 from Integral Molecular for SARS-CoV-2 pseudoviruses or 293T from ATCC for Nipah pseudovirus) in DMEM (phenol red free) containing 10% FBS, and placed in a humidified incubator for 48 or 72 h. Hoechst 33342 (cell-permeable nuclear dye, Thermo) and DRAQ7 (cell impermeable nuclear dye, Thermo) were added and the plate was imaged on a Molecular Devices ImageXpress Micro Confocal Laser at 10x magnification. GFP-positive cells were counted and the percent

GFP positivity was plotted using Prism software (Graphpad). Cytotoxicity was monitored by measuring the ratio of DRAQ7(+)/Hoechst 33342(+) to DRAQ7(-)/Hoechst 33342(+) cells, with no evidence of nonspecific cellular compromise across compound treatments.

## Live-virus assays
**SARS-CoV-2.** SARS-CoV-2 infections were performed in culture and the activity of candidate antivirals was monitored as reported[39]. A549-ACE2 cells were plated in 384-well format and treated for 1–2 h with either a single 4 µM dose or serial dilution (starting at 4 µM, 2 µM, or 100 nM) as indicated of stapled lipopeptides, performed in triplicate, followed by a 48 h incubation with SARS-CoV-2 (Beta B.1.351 [hCoV-19/ South Africa/KRISP-K005325/2020] or Delta B.1.617.2 [hCoV-19/USA/ MA-NEIDL-01399/2021]), at an MOI of ~0.2. Infected cells were then washed, fixed with 10% formalin, removed from containment, rewashed in PBS, immunostained with anti-SARS-CoV-2 nucleocapsid monoclonal antibody (Sino Biological; RRID# AB_2827975; 1:10,000 dilution) followed by anti-Ig secondary antibody (Alexa Fluor 488, Thermo Fisher Scientific; RRID# AB_2576217; 1:1000 dilution), and cell nuclei counterstained with Hoechst 33342. Cells were imaged on a Biotek Cytation 1 automated microscope and analyzed by CellProfiler software (Broad Institute). Infection efficiency was calculated by dividing the number of infected cells by total nuclei and normalized to vehicle-treated controls.

**Respiratory syncytial virus.** A549 cells (ATCC Cat# CCL-185) were plated in 384-well format and treated for 30 min with either a single 2 µM dose or serial dilution as indicated of stapled lipopeptides, followed by addition of GFP-RSV live virus (0.1–0.4 µL virus/well; ViraTree) and incubation for 48–72 h. Infected cells were then washed with PBS and Hoechst 33342 (cell-permeable nuclear dye) and DRAQ7 (cell impermeable nuclear dye) were added, followed by imaging of the plate on a Molecular Devices ImageXpress Micro Confocal Laser at 10x magnification. GFP-positive cells were counted and the percent GFP positivity, as compared to vehicle-treated controls, was plotted using Prism software (Graphpad). Cytotoxicity was monitored by measuring the ratio of DRAQ7(+)/Hoechst 33342(+) to DRAQ7(-)/Hoechst 33342(+) cells, with no evidence of nonspecific cellular compromise across compound treatments.

**Ebola.** Testing for the Ebola virus was performed in a similar manner to that described above for SARS-CoV-2. Briefly, HeLa cells (ATCC Cat# CCL-2) were plated in 384-well format and the following morning a 2-fold serial dilution of peptide starting from 10 µM was applied. The plates were then brought into the BSL4 laboratory and infected with EBOV Mayinga at an MoI of 0.1–0.2. After ~48 h, the plates were fixed in 10% formalin and removed from containment. The plates were then washed and immunostained with a monoclonal anti-EBOV GP antibody (IBT Bioservices; RRID# AB_2754983; 1:4,000 dilution) followed by anti-Ig secondary antibody (Alexa Fluor 488, Thermo Fisher Scientific; RRID# AB_2534088; 1:1000 dilution). Nuclei were counterstained with Hoechst 33342 and analyzed as described above for SARS-CoV-2.

## Design and production of recombinant SARS-CoV-2 5-helix bundle protein
The SARS-CoV-2 5-HB protein incorporates five of the six helices that form the core of the SARS-CoV-2 spike trimer of hairpins. The helices are connected by short peptide linkers as previously designed for the HIV-1 gp41 5-HB[40] and RSV F 5-HB:[19] MQKLIANQFNSAIGKIQDSLSS TASALGKLQDVVNQNAQALNTLVKQLSSGGSGGDISGINASVVNIQKEIDR LNEVAKNLNESLIDLQELGSSGGQKLIANQFNSAIGKIQDSLSSTASALGKL QDVVNQNAQALNTLVKQLSSGGSGGDISGINASVVNIQKEIDRLNEVAKN LNESLIDLQELGSSGGQKLIANQFNSAIGKIQDSLSSTASALGKLQDVVNQ NAQALNTLVKQLSSSSGGHHHHHH. The plasmid (GenScript) was

transformed into *Escherichia coli* BL21 (DE3), grown overnight at 37 °C in Luria broth after induction with 0.1 M isopropyl β-D-thiogalactoside. The cells were collected by centrifugation for 20 min at 5000 x g, resuspended in Buffer A (100 mM NaH$_2$PO$_4$, 20 mM Tris, 8 M urea, pH 7.4), and lysed by agitation at 4 °C overnight. The resulting mixture was clarified by centrifugation (35,000 x g for 30 min) before binding to a nickel-nitrilotriacetate (Ni-NTA) agarose (Qiagen) column at room temperature. After washing with Buffer A (pH 6.3), the 5-HB was eluted with Buffer A (pH 4.5), renatured by diluting (1:4) with PBS (50 mM sodium phosphate, 100 mM NaCl, pH 7.5), and concentrated in a 10-kDa Amicon centricon (diluting and reconcentrating 4 times), yielding ~1 mg/mL protein solution. SDS-PAGE confirmed protein purity (>90%).

## AlphaScreen binding assay
AlphaScreen is a bead-based, non-radioactive Amplified Luminescent Proximity Homogeneous Assay (Alpha) in which laser excitation induces a photosensitizer in the streptavidin donor bead (Perkin Elmer) to convert ambient oxygen into a more excited singlet state. The singlet state oxygen molecules diffuse briefly to react with a thioxene derivative on the anti-His$_6$ acceptor bead (Perkin Elmer) if a complex is formed in sufficient proximity. Here, Biotin-PEG$_2$-labeled peptides (25 nM) were incubated with a serial dilution from 30 nM to 100 pM of His$_6$-labeled SARS-CoV-2 5-HB protein in 1x AlphaLISA HiBlock Buffer (Perkin Elmer) for 1 h. Subsequently, 10 µg/mL of both donor and acceptor beads were added and the plate read at 24 h using an Envision microplate reader (Perkin Elmer). Binding assays were run in quadruplicate, and IC$_{50}$s were calculated by nonlinear regression analysis using Prism software (GraphPad).

## Assessment of peptide homogeneity by size exclusion chromatography
The homogeneity of RQ-01 was assessed by SEC using an Agilent 1260 HPLC equipped with an autosampler, thermostatted column compartment, and UV absorbance detector. RQ-01 was dissolved in sodium phosphate buffer (pH 7.4) containing 150 mM NaCl to achieve a final concentration of 25 µM and, after incubation for 6 h at room temperature, a 10 µL sample was injected onto a Superdex 200 Increase 5/ 150 GL gel permeation column (Cytiva) and eluted in 50 mM sodium phosphate buffer (pH 7.4) containing 150 mM NaCl using a flow rate of 0.2 mL/min.

## Analysis of peptide structure in solution by circular dichroism
The CD spectrum of RQ-01 was recorded on an Aviv Biomedical spectrometer (Model 410) with the temperature set to 20 °C using a Peltier temperature controller, 1-mm path-length cells, and a thermoelectric sample changer with 5-position rotor. The spectrometer scanned in 0.5-nm increments from 190–260 nm with a 0.5-s averaging time (five scans were averaged). RQ-01 was dissolved in 5 mM potassium phosphate (pH 7.5) at a target concentration of 50 µM, with the exact concentration determined by amino acid analysis and used to calculate percent α-helical content, as reported[41].

## Immunofluorescence imaging of RQ-01 in treated cells
A549-ACE2 cells were plated at 3 x 10$^5$ cells/well in 8-well chamber slides (Ibidi, Gräfelfing, Germany). The following morning, Cy5-β-Ala-RQ-01 was added to the cells at a final concentration of 10 µM. After 3–4 h, the medium was removed and cells were fixed in 10% neutral-buffered formalin. The cells were then washed twice in PBS and Hoechst 33342 was added to counterstain the nuclei. Cells were imaged at 100X on a Nikon Ti2 Eclipse microscope.

## Measurement of in vitro peptide stability by LC/MS
Peptides were dissolved in 200 µL of 1 M HCl, 0.1 M NaOH, or 50 mM sodium phosphate buffer (pH 7.4) containing 150 mM NaCl to achieve

a final concentration of 25 µM. Triplicate samples were injected onto an LC/MS (Agilent) at time zero and either 1 h (neutral buffer solution heated to 80 °C) or 12 h (acidic and basic solutions), and the integrated area of the relevant ion (M + 3H/3) or (M + 4H/4) was plotted using Prism software (Graphpad).

## Evaluation of lipopeptide solubility

RQ-01 and SARS$_{HRC}$-PEG$_4$-chol powders were each mixed with 50 mM sodium phosphate buffer, pH 7.0, sonicated for 30 min at room temperature to achieve 15, 30, 60, and 90 mg/mL solutions, and then the glass vials containing the mixtures were visually assessed for lipopeptide solubilization and photographed.

## Measurement of RQ-01 levels in tissues

**Experimental design and treatment.** Male C57BL/6 mice, aged 6-7 weeks, were procured from Charles River Labs and housed in ventilated Innovive IVC cages with food and water available ad libitum. Mice were housed 3 per cage and allowed to acclimate in-house for 1 week prior to the onset of the study (12 h light/dark cycle, ambient temperature 70 ± 2 °F, 50% humidity). Prior to intranasal instillation, mice were anesthetized using inhaled isoflurane. The level of anesthesia was assessed by pedal reflex. Once sufficiently anesthetized, mice were removed from the induction chamber and nasal instillation of RQ-01 was performed immediately. Specifically, anesthetized mice were lightly scuffed and held in a recumbent position at an approximate 45-degree angle; a manual pipette containing 15 µL of stapled lipopeptide test material, delivered in sodium phosphate buffer pH 6.5 with 0.1% w/v hydroxypropyl methylcellulose (HPMC) in deionized water, was placed just outside the animal's right nare, forming a droplet that the animal was allowed to inhale. After inhalation, the animal was held for approximately 10 s to confirm proper delivery. The animal was immediately placed back into the isoflurane chamber to ensure an adequate plane of anesthesia. The process was repeated, alternating nares 3 times for a total delivery of 45 µL of test substance (delivering 1, 3, or 10 mg/kg in 1.8 mL/kg volume). After instillation was complete, the animal was fully monitored and allowed to recover in its home cage. Following compound administration, mice were then sacrificed at set time intervals (n = 3 per dosing level and time point) for blood draw, and nasal turbinate and lung tissue excision and processing for LC/MS analysis (see below). Clinical observations were recorded throughout the study. There were no adverse clinical findings for the duration of the study. All mice were bright, alert, responsive, and displayed normal behavior.

For the correlative hamster study, male Syrian (Golden) hamsters (strain code 049), aged 8–9 weeks, were purchased from Charles River Labs and treated under ketamine-xylazine anesthesia with 200 µL (100 µL/nare) of intranasally instilled RQ-01 (2.5 mg/kg) prepared as above. Hamsters were sacrificed at 4 or 24 h after compound administration (n = 2 per dosing level per time point) for nasal turbinate and lung excision and processing for LC/MS analysis (see below).

**Preparation of tissue and plasma extract samples.** Lung and nasal tissues were processed immediately after harvesting. After weighing the tissues, extraction buffer consisting of 1:1 acetonitrile:H$_2$O containing 0.1% NH$_4$OH was added at a volume of 0.4 mL buffer per 25 mg of nasal turbinate tissue and a volume of 0.5 mL per 200 mg of lung tissue, followed by homogenization using a powered hand homogenizer. The resulting homogenate was collected, added to a new 1.5 mL microfuge tube, and centrifuged at 14,000 rpm (16,000 x g) in a tabletop microfuge at room temperature. The resulting supernatant (extract) was collected and stored at -20 °C until analysis. Blood samples were collected in microtainers containing EDTA. After gently inverting the tubes to ensure mixing with the EDTA coating, the tubes were spun at 8000 rpm (5200 x g) in a tabletop microfuge to separate the plasma from the cell pellet. A 20 µL aliquot of the collected plasma

was added to 100 µL of methanol containing 0.1% NH$_4$OH (plasma extraction buffer) in a 1.5 mL microfuge tube. The samples were completely mixed by vortexing to precipitate protein and then spun at 14,000 rpm (16,000 x g) in a tabletop microfuge at room temperature. The supernatant was collected, added to a new microfuge tube, and stored at −20 °C until analysis.

**LC/MS analysis.** To determine tissue levels of RQ-01, as performed at Charles River Labs, 50 µL of tissue extract sample was transferred to a 2-mL deep well polypropylene plate. After the addition of 50 µL of internal standard ([D4] anandamide at a concentration of 25 ng/mL in 1:1 methanol:acetonitrile) and 200 µL of 1:1 acetonitrile:water containing 0.1% ammonium hydroxide, the tubes were vortexed for 1–2 min. After 10 min centrifugation at 3012 x g, 200 µL of supernatant was transferred to a 2-mL LoBind deep well polypropylene plate. The capped plates were placed in a Thermo CTC PAL autosampler and 20 µL injected into the analytical column. The samples were analyzed on a Waters Xbridge C18 3.5 µm (30 mm x 2.1 mm) column at 45 °C using a gradient with 20 mM ammonium formate in 50:50:0.2 (v:v) water:acetonitrile:formic acid (mobile phase A) and 20 mM ammonium formate in 90:10:0.2 (v:v) acetonitrile:water:formic acid (mobile phase B) as eluents at a constant flow rate of 0.800 mL/min with a total run-time of 2.5 min. The mass spectrometer (API 4000 triple quadrupole) was operated in positive ion mode (ESI + ) with an electrospray voltage of 5500 V at 550 °C. The product ions and collision energies used were m/z 1285 → 369.3 for RQ-01 (CE 45 V) and m/z 352.2 → 66.1 for [D4] anandamide (internal standard) (CE 18 V). The lower limit of quantitation in nasal and lung tissues was 25.0 ng/mL.

For analysis of plasma levels of RQ-01, as performed at Charles River Labs, 25 µL of plasma extract was transferred to a 2-mL deep well polypropylene plate. After addition of 25 µL of internal standard ([D4] anandamide at a concentration of 25 ng/mL in 1:1 methanol:acetonitrile) and 100 µL of methanol containing 0.1% ammonium hydroxide, the tubes were vortexed for 1–2 min. After 10 min centrifugation at 3012 × g, 100 µL of the supernatant was transferred to a 2-mL LoBind deep well polypropylene plate. The capped plates were placed in a Thermo CTC PAL autosampler and 20 µL was injected onto the analytical column. The samples were analyzed on a Waters Xbridge C18 3.5 µm (30 mm × 2.1 mm) column at 45 °C using a gradient with 20 mM ammonium formate in 50:50:0.2 (v:v) water:acetonitrile:formic acid (mobile phase A) and 20 mM ammonium formate in 90:10:0.2 (v:v) acetonitrile:water:formic acid (mobile phase B) as eluents at a constant flow rate of 0.800 mL/min with a total run-time of 2.5 min. The mass spectrometer (API 4000 triple quadrupole) was operated in positive mode ion mode (ESI+) with an electrospray voltage of 5500 V at 550 °C. The product ions and collision energies used were m/z 1285 → 369.3 for RQ-01 (CE 45V) and m/z 352.2 → 66.1 for [D4] anandamide (internal standard) (CE 18 V). The lower limit of quantitation in plasma was 25.0 ng/mL.

## In vivo efficacy testing of RQ-01 in a hamster model of SARS-CoV-2 infection

**Animals.** Syrian (Golden) hamsters (strain code 049) of both sexes were obtained from Charles River Labs and were challenged with SARS-CoV-2 at the age of 64 days (average weight of 140 grams). Hamsters were housed in cages of 4, segregated by sex, under BSL3 containment for the duration of the study (12 h light/dark cycle, ambient temperature 70 ± 2 °F, 50% humidity). Hamsters were challenged with the WA-1 (Wuhan) isolate of SARS-CoV-2 under ketamine-xylazine anesthesia by intranasal instillation of virus, with the following outcomes assessed: (1) daily clinical evaluation and scoring, including body weight; (2) collection of oropharyngeal swabs daily on days 1, 2, 3, and 4 post-challenge; (3) euthanasia and necropsy of all animals 4 days post-challenge (hamsters were euthanized by administering a 2x dose

of ketamine-xylazine followed by cervical dislocation); (4) lung samples fixed in formalin for histopathologic evaluation.

**Virus.** The virus used was Colorado State University (CSU) Vero cell passage 2 (05-Mar-21) of SARS-CoV-2 (WA-01 isolate originally from BEI Resources, [NR-52281, Lot 700033175]). The virus was diluted in PBS, and hamsters were challenged with 100 μL intranasally. Back titration of the inoculum following the challenge revealed a titer of $9.8 \times 10^3$ PFU/100 μL.

**Test sample.** RQ-01 powder was formulated by adding sterile-filtered 0.1% hydroxypropyl methylcellulose in 50 mM sodium phosphate, pH 6.5. The vial was sonicated for ~2 min with no heating and gently swirled to achieve full solubilization. The dosing stock was 2.25 mg/mL to deliver a 3 mg/kg dose to the hamsters with a 100 μL dose per nare.

**Experimental design and treatment.** A total of 48 hamsters were used, with equal numbers of each sex, and assigned to arms ($n = 8$ per arm, $n = 4$ per sex) in accordance with the following treatment groups, with the indicated treatment times relative to the viral challenge: Group 1, RQ-01 at −0.5, 24, 48, 72 h (no viral challenge); Group 2, Vehicle at −0.5, 24, 48, 72 h (viral challenge); Group 3, RQ-01 at −0.5, 12, 24, 48, 72 h (viral challenge); Group 4, RQ-01 at −0.5, 24, 48, 72 h (viral challenge); Group 5, RQ-01 at 12, 24, 48, 72 h (viral challenge); and Group 6, RQ-01 at 24, 48, 72 h (viral challenge). Hamsters were treated at the indicated times by intranasal instillation of 200 μL (100 μL/nare) of peptide solution to obtain an average dose of 3 mg/kg. Treatment was conducted under brief isoflurane anesthesia.

**Oropharyngeal swabs.** OP swabs were collected by rotating a standard polyester swab in the oral cavity and pharynx for approximately 5 s and then breaking off the tip in 1 mL of BA-1/FBS and freezing until assay. After thawing, the samples were vortexed and diluted serially in BA-1 for inoculation onto cells. BA-1 medium is a viral transport medium consisting of Tris-buffered MEM containing 1% BSA, supplemented with 5% fetal bovine serum (FBS) for freezing oral swabs.

**Virus titrations.** A double-overlay plaque assay on Vero cells cultured in 12-well plates was used for virus titrations. Samples of OP fluid were serially diluted in BA-1, and 100 μL of each dilution was inoculated onto each well. Plaques were read 2 and 3 days later (1 and 2 days after the second overlay containing neutral red) and virus titers presented as plaque-forming units (PFU).

**Histopathology.** Formalin-fixed tissues collected for histopathology were processed to obtain standard H&E-stained sections and evaluated in a blinded manner by a veterinary pathologist at Inotiv. Pulmonary histopathology scores were an aggregate of scores for overall lesion extent, alveolitis, bronchitis, pneumocyte hyperplasia, and interstitial inflammation.

**Statistical analyses.** One-way ANOVA with Dunnett's Multiple Comparisons Test to vehicle was used to analyze the data. In the case of virus titrations, the data were $\log_{10}$-transformed prior to ANOVA analysis.

### In vivo efficacy testing of RQ-01 in a hamster model of SARS-CoV-2 transmission

**Animals.** Syrian (Golden) hamsters (strain code 049) were obtained from Charles River Labs, housed under BSL3 containment (12 h light/dark cycle, ambient temperature $70 \pm 2$ °F, 50% humidity), and were 65 days of age with an average weight of 148 grams at the onset of the experiment. Only males were used to minimize fighting when animals were mixed. The donor hamsters were challenged with the WA-01 (Wuhan) isolate of SARS-CoV-2 (as described above) under ketamine-xylazine anesthesia by intranasal instillation of 100 μL of a back-titrated dose of $1.1 \times 10^4$ PFU on Day 0. One day after infection of the donors, each animal was transferred to a different cage containing two sentinel hamsters (Day 1), and one day after that, each hamster was separated and housed individually in a fresh cage (Day 2). The following outcomes were assessed: (1) daily clinical evaluation and scoring, including body weight; (2) collection of oropharyngeal swabs; (3) euthanasia and necropsy of all animals on Day 5 (hamsters were euthanized by administering a 2x dose of ketamine-xylazine followed by cervical dislocation); (4) preparation of lung homogenates for virus titrations.

**Virus.** The virus used was Colorado State University (CSU) Vero cell passage 2 (05-Mar-21) of SARS-CoV-2 (WA-01 isolate originally from BEI Resources, [NR-52281, Lot 700033175]). The virus was diluted in PBS, and hamsters were challenged with 100 μL intranasally. Back titration of the inoculum following the challenge revealed a titer of $1.1 \times 10^4$ PFU/100 μL.

**Test sample.** RQ-01 powder was formulated by adding sterile-filtered 0.1% hydroxypropyl methylcellulose in 50 mM sodium phosphate, pH 6.5. The vial was sonicated for ~2 min with no heating and gently swirled to achieve full solubilization. The dosing stock was 2.25 mg/mL to deliver a 3 mg/kg dose to the hamsters with a 100 μL dose per nare.

**Experimental design and treatment.** A total of 36 male hamsters were used, with 12 serving as donors (Group 1) and three cohorts of eight as sentinels (Groups 2–4). The sentinels received the following treatments, at times relative to mixing with donor hamsters on Day 1: Group 2, vehicle at 24 h (Day 2), 48 h (Day 3), 72 h (Day 4); Group 3, RQ-01 at −0.5 h (Day 1), 24 h (Day 2), 48 h (Day 3), 72 h (Day 4); Group 4, RQ-01 at 24 h (Day 2), 48 h (Day 3), 72 h (Day 4). Hamsters were treated at the indicated times by intranasal instillation of 200 μL (100 μL/nare) of peptide solution to obtain an average dose of 3 mg/kg. Treatment was conducted under brief isoflurane anesthesia.

**Oropharyngeal swabs.** OP swabs were collected by rotating a standard polyester swab in the oral cavity and pharynx for approximately 5 s and then breaking off the tip in 1 mL of BA-1/FBS and freezing until assay. After thawing, the samples were vortexed and diluted serially in BA-1 for inoculation onto cells. BA-1 medium is a viral transport medium consisting of Tris-buffered MEM containing 1% BSA, supplemented with 5% fetal bovine serum (FBS) for freezing oral swabs.

**Virus titrations.** A double-overlay plaque assay on Vero cells cultured in 12-well plates (OP swabs) or 6-well plates (lung homogenates) was used for virus titrations. Samples were serially diluted in BA-1, and 100 μL of each dilution was inoculated onto each well. Plaques were read 2 and 3 days later (1 and 2 days after the second overlay containing neutral red) and virus titers presented as plaque-forming units (PFU).

**Statistical analyses.** One-way ANOVA with Dunnett's Multiple Comparisons Test to the vehicle was used to analyze the data. In the case of virus titrations, the data were $\log_{10}$-transformed prior to ANOVA analysis.

### Statistical methods
Prism 10 software (GraphPad) was used for data analyses, including calculating mean, SD, and SEM values and performing one-way ANOVA with Dunnett's Multiple Comparisons Test.

### Reporting summary
Further information on research design is available in the Nature Portfolio Reporting Summary linked to this article.

## Data availability

Protein Data Bank entries with identification numbers "7TIK", "1G2C", "1EBO", and "1WP7" were used in the course of this study. All other data supporting the findings of this study are available within the article and its supplementary files. Any additional requests for information can be directed to, and will be fulfilled by, the corresponding author. Source data are provided in this paper.

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

## Acknowledgements

We thank E. Smith for figure preparation, C. Jonnson for helpful discussions, and the pre-clinical services program offered by the National Institute of Allergy and Infectious Diseases for supporting the in vivo testing of RQ-01 in the hamster model of SARS-CoV-2

infection in the laboratory of Dr. Richard Bowen at Colorado State University. This work was also supported in part by a grant from the Massachusetts Consortium on Pathogen Readiness and NIH grant UC7AI095321 to R.A.D.

## Author contributions

G.H.B. and L.D.W. conceived of and designed the study. G.H.B. and B.M.M. synthesized the stapled lipopeptides. G.H.B. performed all biochemical assays. G.H.B. and M.G. conducted the pseudovirus (SARS-CoV-2, Nipah) and RSV assays. J.J.P., C.D.O., and R.A.D. performed live virus (SARS-CoV-2, Ebola) assays. W.Z., N.B., and P. D.-J. conducted RQ-01 formulation and pharmacokinetic studies. R.A.B. performed the in vivo efficacy studies. D.S.N. performed statistical analyses. All authors analyzed data and contributed to and reviewed the manuscript, which was written by L.D.W.

## Competing interests

G.H.B. is a consultant for and shareholder in Red Queen Therapeutics, P.D.-J. is an employee of and shareholder in Red Queen Therapeutics, and L.D.W. is a scientific co-founder of and consultant for Red Queen Therapeutics. The remaining authors declare no competing interests.
