## [Peer review file · Nature Communications]

REVIEWER COMMENTS

Reviewer #1 (Remarks to the Author):

In this work, the authors designed and synthesized a series of stapled lipopeptide inhibitors of SARS-CoV-2 through identification of the best sites for hydrocarbon-stapling within HR2 (SAH-HR2-D-PEG4-TC), which can effectively inhibit infection of multiple SARS-CoV-2 variants. Subsequently, the authors optimized the linker length between lipid moiety and peptide, and developed the stapled inhibitor (RQ-01) as the final candidate, which showed the potent in-vitro and in-vivo inhibitory activities against SARS-CoV-2 infection. The authors also applied the stapled lipopeptide platform for the development of entry inhibitors of Respiratory Syncytial, Ebola, or Nipah viruses.

Specific comments:

1. Numerous reports have shown that HCoV S-HR2-derived lipopeptides exhibit excellent broad-spectrum antiviral activity with IC₅₀ of 0.3~0.5 nM for inhibiting cell-cell fusion and 0.9~4 nM for inhibiting pseudovirus infection (Antiviral Res. 2022;208:105445; Emerg Microbes Infect. 2022;11:1819-1827), high stability (Acta Pharm Sin B. 2022;12):1652-1661), and good in vivo protectivity in animal models (Science, 2021, 371:1379-1382). The stapled lipopeptides reported in this manuscript did not show significantly better antiviral activity and stability than these reported lipopeptides, while the production cost is expected to be higher than lipopeptides. Therefore, what is the advantages to develop the stapled lipopeptides compared with the unstapled lipopeptides reported?
2. The authors reported that the best stapling sites in staple D, However, HR2-D-PEG4-TC only showed similar potency to that of HR2-PEG4-TC, suggesting that the introduction of these stapling sites may have little contribution on its antiviral efficacy. The authors also found that the optimized linker is PEG lengths of 8 or greater. However, this is also a regular linker, which has been reported by other teams (Emerg Microbes Infect., 2022, 11:1819-1827; Emerg Microbes Infect., 2021, 10: 1227-1240) .
3. What is the specific linker of RQ-01? Why was thiocholesterol (TC) chosen as the modified lipid molecule, instead of the previously reported cholesterol. What is the advantage of this linker?
4. In Figure 6, authors reported several stapled lipopeptides, including SLI-RSV, SLI-EBOV, and SLI-NIPV with potent antiviral activity. However, their specific sequences and linkers were not described.
5. The authors stated that stapling strikingly contributes to the stability of peptide. To better support this conclusion, the half-life (t_{1/2}) of these stapled and unstapled lipopeptides should be provided and compared.
6. Some unstapled HR2-derived lipopeptides, such as, SARSHRC-PEG24-chol (Science, 2021, 371:1379-1382), have shown good nasal bioavailability with in vivo antiviral protection, but have not shown oral bioavailability. If authors could show the oral bioavailability of RQ-01, this stapled lipopeptide would have better clinical application potential than the unstapled lipopeptide.
7. The lipopeptides reported show low cytotoxicity. How is about the cytotoxicity of RQ-01, SLI-RSV, SLI-EBOV, and SLI-NIPV?
8. In this study, several animal experiments are performed, while the animal ethics approval number could not be found in this manuscript.

Reviewer #2 (Remarks to the Author):

In this manuscript, Bird and colleagues build upon prior efforts using a stapled lipopeptide platform to show that this antiviral approach can inhibit SARS-CoV-2 viral entry and reduce morbidity, viral titers, and histopathological alterations in hamsters. There is a continued need for pre- and post-exposure therapeutics that can exhibit efficacy against a broad range of viral pathogens, and studies like this, which employ novel technologies that target conserved mechanisms of infection across multiple virus families, are of high interest to the field. While experiments presented in the manuscript appear generally sound, there are some questions pertaining to the in vivo experiments, and the discussion

section is lacking sufficient context with the rest of the field.

Major comments:

-while the introduction is well-referenced (23 cited papers), there are only 29 references total in this manuscript, which is rather low for a full-length article of this complexity. The discussion appears very under-referenced and brief relative to the introduction and other sections. There are several areas presented throughout the results (including strain-specific results obtained in pseudoviral infectivity assays, how other antiviral or therapeutic approaches have performed relative to results shown here in the hamster model in both pathogenicity and transmissibility settings, relative efficacy of other antiviral or therapeutic approaches which have broad-spectrum applicability against multiple virus families, etc) which all warrant contextualization with the literature yet are lacking here.

-Authors performed tissue pharmacokinetic studies in mice, but assessed efficacy against SARS-CoV-2 in hamsters. Why did the authors not perform PK evaluations in the intended viral challenge model species? How translatable are results from mice regarding dosing and tissue concentration over time between these two species? Furthermore, it's unclear why authors include a mix of frequencies of RQ-01 administration to the animals (in some groups, intervals are 12hr apart whereas in others the interval is 24hr, and in the transmission setup there is a 72hr interval in one of the groups) – if Figure 4 shows comparable tissue maintenance of this drug over 24hr, why include 12-hr administrations? And how long beyond the 24hr interval shown in Figure 4 is the compound still detected? Results presented in Figure 4 would be much more useful to the study overall if it included data from the hamster model showing how long the 3 mg/kg dose was detected from initial administration until clearance from tissue in both the upper and lower respiratory tract.

-Authors show lung homogenate viral titers (supplemental figure 4d) and in vehicle or drug-treated contact animals, and lung histopathology scores in directly inoculated animals, but do not present any pharmacokinetic information regarding if the 200ul total dose of drug administered to animals reaches the lungs (and if so, how long it remains efficacious). Understanding if the drug is reaching the lower respiratory tract (and if so how long it remains efficacious) would improve context for these results. Did the authors determine viral titers in the lungs shown in figures 5e and f like they report in supplemental figure 4?

Minor comments:

-abstract, change "...blocks infection by THE spectrum..." to "...blocks infection by A spectrum" to better reflect the continuous evolution of SARS-CoV-2 viruses.

-Figure 1d-h x axis shows "normalized percent infected cells" but does not specify what these values were actually normalized to (vehicle treated controls?), please specify this in the legend.

-Figure 1d-h, please define "WT" on the left y axis of these figures. Is this to denote the unstapled control peptide? What is the gray shading meant to represent in these figures? What threshold must be met by results for the color of the dots to be red vs black? Please update the legend with all of these missing information.

-Figure 1h, why do the authors feel that so many more stapled constructs tested exhibited higher potency against the Beta SARS-CoV-2 strain relative to the other psueuoviruses examined? This is pointed out in the results but a reason is not discussed.

-Figure 3 panels g-i I believe are mislabeled and should read f-h.

-for data from Supplementary Figure 4, suggest changing "infected transmitter" to "infected donor" to better align with conventional description of donor animals employed in transmission studies in the

literature.

-Supplemental Figure 4d is missing a lower range on the left y axis, so the reader cannot tell what the titer is being shown.

Reviewer #3 (Remarks to the Author):

In this manuscript, Bird et al & Walensky describe the development of lipidated, stapled peptides to block SARS-2 infection by blocking full spike protein fusion with the cell membrane. This is accomplished by the helical peptides binding to the initial helical bundle that the virus inserts into the membrane, preventing further conformational change at the maturation to the full, infection competent bundle (Figure 2a). In cellular and in vivo tests in hamsters, intranasal dosing of 3mg/kg of the peptides reduces infection and re-transmission.

Overall, I found this a compelling study, certainly topical, that will interest many of the Nat. Communications readership. There are three points that should be addressed to strengthen the manuscript:

1. The PK study seemed superficial. The authors dose the hamsters intranasally, and then examine the distribution of the drug over time in nasal tissue homogenites. They do not report the distribution of the stapled peptides in any other organ. I would think that knowing how they distribute into the plasma, the upper airways, and the lungs would be pertinent to their potential as therapeutics. If the authors believe that having them stay exclusively in the nares is beneficial, or sufficient, they should say so--but it'd still be worthwhile to see their distribution elsewhere, especially if they are used to treat individuals who are already infected, as I expect would be the typical use case of an antiviral drug.

2. The authors write that an anti-HIV peptide drug, enfuvirtide, suffered from "its high cost, lack of oral bioavailability, and poor in vivo stability," and presumably their leads are superior on those counts. But I'm not sure they really demonstrated that. Is there reason to believe that their lipidated, stapled peptide will be much cheaper than enfuvirtide? I may have missed it, but do they believe their molecule will be orally bio-available or systemically stable (I saw no in vivo time courses, except in the nares themselves). Is stability in the nares sufficient? Maybe it is, but if so perhaps they could make this point and justify it.

3. A small point is that I was confused by swab titer experiments in Figure 5b and 5d. The y-axis is (Log₁₀ PFU), but it is itself plotted on a log scale. I think it should be a linear scale representing log₁₀ units (i.e., a value of 1 would be 10, a value of 2 would be 100..., with the distance between 1, 2, 3... being equal). But perhaps I'm misinterpreting it. So, for instance, do the PFUs for untreated vs. treated hamsters drop from about 3000 to about 1 (classes 2 to 3) in Fig. 5b, or do they drop from 3.5 to 1? This could be clarified.

Notwithstanding these critiques, I thought this was overall a strong paper; addressing the PK concerns, especially, would strengthen it.

Bird et al.
Response to Review

Reviewer #1:

In this work, the authors designed and synthesized a series of stapled lipopeptide inhibitors of SARS-CoV-2 through identification of the best sites for hydrocarbon-stapling within HR2 (SAH-HR2-D-PEG4-TC), which can effectively inhibit infection of multiple SARS-CoV-2 variants. Subsequently, the authors optimized the linker length between lipid moiety and peptide, and developed the stapled inhibitor (RQ-01) as the final candidate, which showed the potent in-vitro and in-vivo inhibitory activities against SARS-CoV-2 infection. The authors also applied the stapled lipopeptide platform for the development of entry inhibitors of Respiratory Syncytial, Ebola, or Nipah viruses.

Specific comments:

1. Numerous reports have shown that HCoV S-HR2-derived lipopeptides exhibit excellent broad-spectrum antiviral activity with IC₅₀ of 0.3~0.5 nM for inhibiting cell-cell fusion and 0.9~4 nM for inhibiting pseudovirus infection (Antiviral Res. 2022;208:105445; Emerg Microbes Infect. 2022;11:1819-1827), high stability (Acta Pharm Sin B. 2022;12):1652-1661), and good in vivo protectivity in animal models (Science, 2021, 371:1379-1382). The stapled lipopeptides reported in this manuscript did not show significantly better antiviral activity and stability than these reported lipopeptides, while the production cost is expected to be higher than lipopeptides. Therefore, what is the advantages to develop the stapled lipopeptides compared with the unstapled lipopeptides reported?

2. The authors reported that the best stapling sites in staple D, However, HR2-D-PEG4-TC only showed similar potency to that of HR2-PEG4-TC, suggesting that the introduction of these stapling sites may have little contribution on its antiviral efficacy. The authors also found that the optimized linker is PEG lengths of 8 or greater. However, this is also a regular linker, which has been reported by other teams (Emerg Microbes Infect., 2022, 11:1819-1827; Emerg Microbes Infect., 2021, 10: 1227-1240).

1 and 2: A key attribute of all-hydrocarbon stapling of peptides is recapitulation and fortification of native alpha-helical secondary structure, thereby conferring structural stability while maintaining, and often enhancing, bioactivity. Indeed, this advantage was initially documented in the field of viral fusion inhibitors, whereby we generated stapled HIV-1 and RSV HR2 domains, which were remarkably more structured and stabilized than their unstapled counterparts. The beneficial attributes of stapling have extended to a series of anti-cancer peptides, one of which progressed to Phase 1 and 2 clinical testing in adults and children and another just entered Phase 1 trials. In the revised manuscript, we now better highlight the documented advantages of stapling for helical peptides in general, and the HR2 peptides in particular, and also elaborate on our two foundational HR2 references, Bird et al. PNAS, 2010 and Bird et al. JCI, 2014 in the discussion section.

In the current study, we demonstrate that harnessing the benefits of stapling in the context of the SARS-CoV-2 sequence involved thorough “staple scanning”, which revealed that one staple position in particular faithfully recapitulated biological activity in pseudoviral assays compared to the corresponding unstapled peptide (Figure 1d-h, 2g-h). Further optimization, including appending and iterating a linker-lipid, resulted in potency enhancement in pseudoviral and live virus assays (Figure 2a-f, 2h-i). When comparing the optimized constructs to their unstapled counterparts, several advantages became apparent:

1. increased target binding activity, with stapling conferring a 7-fold increase in affinity for the 5-helix bundle compared to the corresponding unstapled peptide (new Supplementary Fig. 6);
2. up to 2-fold enhancement in antiviral potency in pseudoviral and live virus assays of the D-stapled hit that emerged from the initial screen (Fig. 1i, new Supplementary Figure 3, Fig. 2g-h) and up to 23-fold improvement of the optimized RQ-01 construct compared to an unstapled analog with otherwise similar amino acid sequence, linker length, and lipid (Fig. 3c-e);
3. strikingly enhanced stability in the face of acid, base, and heat exposure for the stapled (RQ-01) vs. unstapled (SP4C) peptide (Fig. 3g-i);
4. markedly improved solubility, which was a notable limitation in advancing unstapled lipopeptides to preclinical testing in small animal models (e.g., DeVries et al. *Science*, 2021). Indeed, we show that RQ-01 is soluble across a 15-90 mg/mL concentration range in aqueous buffer (without any organic solvent), but the unstapled SP4C peptide exhibits no solubility (see new Supplementary Fig. 7). The notable solubility of our stapled lipopeptides, as facilitated by pre-organizing the hydrophobic and hydrophilic faces of the peptide, is a major advantage that has contributed to successful advancement of our lead compound, RQ-01, to clinical testing; and
5. with respect to cost of goods and manufacturing, another key advance is adapting all synthetic steps, including lipidation (otherwise performed in solution in all previously published reports), to the solid phase (Supplementary Fig. 1), which significantly streamlined the cost of production and purification for upscaled drug production. Of note, three companies have now advanced stapled peptides to clinical testing for various disease indications, highlighting the practicality of upscaling synthesis and purification to generate clinical grade material in support of human testing.

In the revised discussion, we further contextualize the above-described stapled peptide advantages within the published literature, incorporating the helpful references listed by the Reviewer. We agree with the Reviewer that the above-described experimental and textual revisions were important for us to further clarify and emphasize how stapling is a beneficial attribute of our stapled lipopeptide platform, which combines stapling, lipidation, and thorough iteration of linker length.

3. What is the specific linker of RQ-01? Why was thiocholesterol (TC) chosen as the modified lipid molecule, instead of the previously reported cholesterol. What is the advantage of this linker?

4. In Figure 6, authors reported several stapled lipopeptides, including SLI-RSV, SLI-EBOV, and SLI-NIPV with potent antiviral activity. However, their specific sequences and linkers were not described.

3 and 4: In the revised manuscript, we demonstrate that our stapled lipopeptides can be generated either as thiocholesterol or cholesterol derivatives, using the same solid phase synthetic approach, thus broadening the versatility of our method for generating peptides with either version of lipidated species (see revised Supplementary Fig. 1). Per the Reviewer's recommendation, we further include as new Supplementary Table 1, a comprehensive listing of all lipopeptide constructs synthesized for the study, including peptide sequences, linker compositions, and lipid moieties. We explain that for preclinical testing and translation we opted for incorporating cholesterol over thiocholesterol because of the substantially lower cost of the starting materials to generate 2-(cholesterol)acetic acid.

5. The authors stated that stapling strikingly contributes to the stability of peptide. To better support this conclusion, the half-life ($t_{1/2}$) of these stapled and unstapled lipopeptides should be provided and compared.

In Fig. 3, we demonstrate the striking differences in the stapled vs. unstapled peptides in response to acid, base, and heat. In each case, the unstapled peptide is decomposed to 6-30% of initial levels by 12 h, whereas the stapled peptide is fully (100%) preserved (precluding a calculation of half-life in this context). These data are consistent with our prior work with stapled HIV-1 and RSV HR2 domain peptides and stapled GLP-1 peptides, which exhibit half-lives of up to 2-4 orders of magnitude greater than their unstapled counterparts. We further provide a series of PK analyses (Fig. 4, S8) demonstrating that RQ-01 treatment, delivered intranasally, provides sustained nasal tissue exposure ($>100\text{-}2000\times$ IC₉₀ value, as measured over 0-24 h), allowing for a convenient q24h dosing regimen *in vivo*.

6. Some unstapled HR2-derived lipopeptides, such as, SARSHRC-PEG24-*chol* (*Science*, 2021, 371:1379-1382), have shown good nasal bioavailability with *in vivo* antiviral protection, but have not shown oral bioavailability. If authors could show the oral bioavailability of RQ-01, this stapled lipopeptide would have better clinical application potential than the unstapled lipopeptide.

In our original work, we generated stapled HIV-1 HR2 peptides and demonstrated oral bioavailability because the intent was systemic treatment for blood-borne infection. Here, we aim to develop a topical treatment for the prevention and symptomatic treatment of respiratory infection, and thus intentionally focused our translational efforts on intranasal rather than systemic administration.

As described above, we further find that our lead stapled lipopeptide exhibits notably enhanced stability compared to an unstapled analog in the face of acid, base, and heat exposure – attributes also beneficial for short and long-term storage. These features obviate the need for maintaining a cold chain and are also important for stockpiling. We further find that the stapled construct is strikingly more soluble than the unstapled versions reported by DeVries et al. (*Science*, 2021), who document their challenges with dose-limiting solubilization of their constructs in aqueous (and even DMSO-containing) buffers.

7. The lipopeptides reported show low cytotoxicity. How is about the cytotoxicity of RQ-01, SLI-RSV, SLI-EBOV, and SLI-NIPV?

For this revision, we now include cell viability experiments for each of our lead lipopeptide constructs and show no cytotoxicity (new Supplementary Fig. S5).

8. In this study, several animal experiments are performed, while the animal ethics approval number could not be found in this manuscript.

The animal ethics approval numbers are listed in the final sentence of the methods sections for the *in vivo* PK and efficacy studies.

We thank the Reviewer for motivating us to conduct a series of additional experiments and provide further discussion to highlight the contribution of stapling to our stapled lipopeptide platform. We believe that the combined features of our compositions represent an important advance worthy of dissemination, so that natural peptides, like

SARS-CoV-2 HR2 domains, can at long last yield a target product profile compatible with advancement to human testing.

Reviewer #2

In this manuscript, Bird and colleagues build upon prior efforts using a stapled lipopeptide platform to show that this antiviral approach can inhibit SARS-CoV-2 viral entry and reduce morbidity, viral titers, and histopathological alterations in hamsters. There is a continued need for pre- and post-exposure therapeutics that can exhibit efficacy against a broad range of viral pathogens, and studies like this, which employ novel technologies that target conserved mechanisms of infection across multiple virus families, are of high interest to the field. While experiments presented in the manuscript appear generally sound, there are some questions pertaining to the in vivo experiments, and the discussion section is lacking sufficient context with the rest of the field.

We are very appreciative of the Reviewer's endorsement of advancing novel technologies - such as the stapled lipopeptide platform - to address the unmet need of pre- and post-exposure therapeutic modalities for a broad range of viral pathogens. We thank the Reviewer for characterizing our work as "sound" and "of high interest to the field". We have now incorporated the Reviewer's recommendations for improving our manuscript, as detailed below.

Major comments:

-while the introduction is well-referenced (23 cited papers), there are only 29 references total in this manuscript, which is rather low for a full-length article of this complexity. The discussion appears very under-referenced and brief relative to the introduction and other sections. There are several areas presented throughout the results (including strain-specific results obtained in pseudoviral infectivity assays, how other antiviral or therapeutic approaches have performed relative to results shown here in the hamster model in both pathogenicity and transmissibility settings, relative efficacy of other antiviral or therapeutic approaches which have broad-spectrum applicability against multiple virus families, etc) which all warrant contextualization with the literature yet are lacking here.

We have taken this recommendation to heart and have now expanded both our discussion text (1018 words up from 728) and discussion references (30 references up from 4) to better contextualize our work with respect to the published literature in the field.

-Authors performed tissue pharmacokinetic studies in mice, but assessed efficacy against SARS-CoV-2 in hamsters. Why did the authors not perform PK evaluations in the intended viral challenge model species? How translatable are results from mice regarding dosing and tissue concentration over time between these two species? Furthermore, it's unclear why authors include a mix of frequencies of RQ-01 administration to the animals (in some groups, intervals are 12hr apart whereas in others the interval is 24hr, and in the transmission setup there is a 72hr interval in one of the groups) – if Figure 4 shows comparable tissue maintenance of this drug over 24hr, why include 12-hr administrations? And how long beyond the 24hr interval shown in Figure 4 is the compound still detected? Results presented in Figure 4 would be much more useful to the study overall if it included data from the hamster model showing how long the 3 mg/kg dose was detected from initial administration until clearance from tissue in both the upper and lower respiratory tract.

As a practical matter, the laboratory we used to perform the PK evaluations were not certified to conduct hamster studies, which is why we conducted our PK studies in mice. Then, upon identifying a site for *in vivo* efficacy testing (CSU), our virology collaborator (Dr. Richard Bowen), which employed hamster models, spot-checked the dosing we recommended based on the murine work and found encouraging/reassuring tissue levels in hamsters as well. We now include these hamster PK data as Supplementary Fig. S8b. To further ensure that we didn't miss the mark upon translating our dosing recommendation from mouse to hamster, we included arms with extra dosing time points (12 h, 24 h, etc.), just to be on the safe side. Ultimately, the PK studies performed in mice were very informative in arriving at effective doses for treating the hamsters. With respect to the Reviewer's question about tissue levels of RQ-01 beyond 24 hours, we now include additional murine PK data that monitored nasal and tissue levels out to 48 hours (see Supplementary Fig. 4b). In the revised manuscript, we correspondingly expand our discussion of the formulations, PK methods, and above-described rationale.

-Authors show lung homogenate viral titers (supplemental figure 4d) and in vehicle or drug-treated contact animals, and lung histopathology scores in directly inoculated animals, but do not present any pharmacokinetic information regarding if the 200ul total dose of drug administered to animals reaches the lungs (and if so, how long it remains efficacious). Understanding if the drug is reaching the lower respiratory tract (and if so how long it remains efficacious) would improve context for these results. Did the authors determine viral titers in the lungs shown in figures 5e and f like they report in supplemental figure 4?

As described above, we now include the requested PK data related to lung tissue levels in new Figure 4b and new Supplementary Figures 8a-b.

The two *in vivo* studies were performed at different times, each designed to have a health monitoring outcome (weight), a nasal tissue assessment (OP swab), and a lung tissue assessment, with the first study (efficacy) focusing on lung histopathology and the second study (transmission) on lung viral titers.

Minor comments:

-abstract, change "...blocks infection by THE spectrum..." to "...blocks infection by A spectrum" to better reflect the continuous evolution of SARS-CoV-2 viruses.

The textual change was made as recommended.

-Figure 1d-h x axis shows "normalized percent infected cells" but does not specify what these values were actually normalized to (vehicle treated controls?), please specify this in the legend.

The normalization is now defined in the legend as anticipated by the Reviewer: "The data are normalized to percent infected cells treated with vehicle control."

-Figure 1d-h, please define "WT" on the left y axis of these figures. Is this to denote the unstapled control peptide? What is the gray shading meant to represent in these figures? What threshold must be met by results for the color of the dots to be red vs black? Please update the legend with all of these missing information.

The legend was updated and clarified with the requested information as suggested: "WT, unstapled lipopeptide bearing the indicated wild-type HR2 domain sequence."

-Figure 1h, why do the authors feel that so many more stapled constructs tested exhibited higher potency against the Beta SARS-CoV-2 strain relative to the other pseudoviruses examined? This is pointed out in the results but a reason is not discussed.

We clarify in the legend and main text that the screening doses for pseudoviral assays were 250 or 500 nM, but was higher at 4 μ M for the live virus assay, explaining why there were more “hits” in the live virus screening assay. We further explain that these additional hits observed in the live virus assay also scored as relatively more potent than others in the pseudoviral assays as well.

-Figure 3 panels g-i I believe are mislabeled and should read f-h.

We have now made this correction in figure labeling.

-for data from Supplementary Figure 4, suggest changing “infected transmitter” to “infected donor” to better align with conventional description of donor animals employed in transmission studies in the literature.

We made the textual change to “infected donor” as suggested.

-Supplemental Figure 4d is missing a lower range on the left y axis, so the reader cannot tell what the titer is being shown.

We adjusted the labeling of the y axis to resolve this.

Reviewer #3

In this manuscript, Bird et al & Walensky describe the development of lipidated, stapled peptides to block SARS-2 infection by blocking full spike protein fusion with the cell membrane. This is accomplished by the helical peptides binding to the initial helical bundle that the virus inserts into the membrane, preventing further conformational change at the maturation to the full, infection competent bundle (Figure 2a). In cellular and in in vivo tests in hamsters, intranasal dosing of 3mg/kg of the peptides reduces infection and re-transmission.

Overall, I found this a compelling study, certainly topical, that will interest many of the Nat. Communications readership. There are three points that should be addressed to strengthen the manuscript:

We are very appreciative of the Reviewer characterizing our study as “compelling,” “certainly topical,” and “will interest many of the Nat. Communications readership”. We have now performed all of the suggested revision experiments in accordance with the Reviewer’s recommendations, as summarized below.

1. The PK study seemed superficial. The authors dose the hamsters intranasally, and then examine the distribution of the drug over time in nasal tissue homogenites. They do not report the distribution of the stapled peptides in any other organ. I would think that knowing how they distribute into the plasma, the upper airways, and the lungs would be pertinent to their potential as therapeutics. If the authors believe that having them stay exclusively in the nares is beneficial, or sufficient, they should say so--but it'd still be worthwhile to see their distribution

elsewhere, especially if they are used to treat individuals who are already infected, as I expect would be the typical use case of an antiviral drug.

We now include further PK studies, measuring drug levels in lung tissue and blood, in addition to nasal tissue. These new PK data are presented in new Fig. 4b and new Supplementary Figures 8a and 8b.

2. The authors write that an anti-HIV peptide drug, enfuvirtide, suffered from "its high cost, lack of oral bioavailability, and poor in vivo stability," and presumably their leads are superior on those counts. But I'm not sure they really demonstrated that. Is there reason to believe that their lipidated, stapled peptide will be much cheaper than enfuvirtide? I may have missed it, but do they believe their molecule will be orally bio-available or systemically stable (I saw no in vivo time courses, except in the nares themselves). Is stability in the nares sufficient? Maybe it is, but if so perhaps they could make this point and justify it.

We clarify in the revised manuscript that our mention of enfuvirtide's lack of oral bioavailability was made in reference to the stapled HIV-1 HR2, which exhibited measurable oral bioavailability (Bird et al PNAS, 2010), highlighting an advantage of stapling. In the context of SARS-CoV-2, we are instead focused on developing a topical agent for intranasal delivery. We find relatively little systemic absorption upon intranasal treatment, suggesting that the observed efficacy derives from topical rather than systemic activity. We now include a comparison of RQ-01 levels in nasal tissue, lung tissue, and plasma after intranasal administration of 1, 3, and 10 mg/kg single doses (see new Supplementary Fig. S8a).

3. A small point is that I was confused by swab titer experiments in Figure 5b and 5d. The y-axis is (Log₁₀ PFU), but it is itself plotted on a log scale. I think it should be a linear scale representing log₁₀ units (i.e., a value of 1 would be 10, a value of 2 would be 100..., with the distance between 1, 2, 3... being equal). But perhaps I'm misinterpreting it. So, for instance, do the PFUs for untreated vs. treated hamsters drop from about 3000 to about 1 (classes 2 to 3) in Fig. 5b, or do they drop from 3.5 to 1? This could be clarified.

We revised the y-axis in accordance with the Reviewer's suggestion to avoid any confusion on this important point. Indeed, the drop from vehicle (arm 2) to treatment regimens 3 and 4 is exponential (from 3500 to 10-20 PFUs).

Notwithstanding these critiques, I thought this was overall a strong paper; addressing the PK concerns, especially, would strengthen it.

We are very thankful for the Reviewer's endorsement of our work as "overall a strong paper" and for motivating us to include additional PK analyses to further strengthen the paper.

We want to thank the Reviewers for their time and effort in carefully reviewing our manuscript. We hope that by performing all of the suggested experiments, as now included as Figures 4b, Supplementary Figures S3, S5a-e, S6, S7, S8a-b, and Supplementary Table 1, in addition to the recommended textual changes, you will find our manuscript suitable for publication in *Nature Communications*.

REVIEWERS' COMMENTS

Reviewer #1 (Remarks to the Author):

The authors have satisfactorily addressed all my concerns and revised the manuscript accordingly. Therefore, this manuscript meets the high-quality standards of Nature Communication.

Reviewer #2 (Remarks to the Author):

Authors have addressed all comments raised during peer review; no additional comments.

Reviewer #3 (Remarks to the Author):

The authors have done PK experiments to address my concerns, and I commend them for their further effort. I certainly do not think they need to do further experiments, but I would suggest the following textual modifications:

All three of the reviewers had concerns about the PK and in vivo stability of the new stapled peptides, and while the authors have partly addressed that, key issues remain uncertain. For instance, as they note in their cover letter, the gross amount of compound that is in the general circulation is actually fairly low, probably too low for anti-viral efficacy over the course of a day.

Probably more importantly still, the exposures that they measure--the concentrations in the nares, lung, and the blood--are all gross amounts, not fraction unbound (F_u). It is the latter that is crucial for efficacy, and it will be lower, sometimes much lower, than the gross amount that they measure (e.g., it's not conceivable that the fraction unbound is only 5%, or even less, of the gross amount that they measure).

Given that the blood and even the lung C_{max} values drop from those in the nares (by 100- and 10-fold, respectively), and that the fraction unbound will be lower still, maybe log-orders lower, I recommend that the authors add a caveat to the Discussion, to bring this to the attention of readership. After all, they motivate the study by writing that stapled, lipidated peptides can address what are essentially PK issues. It seems to me that several of these remain unaddressed, or at least uncertain, from their study. Bringing this to the attention of their readers in the Discussion will strengthen the paper, because readers will appreciate their candor.

This is just a recommendation, I leave it to the Editors and the authors discretion as to whether they want to do this. I don't need to see this manuscript again and do support publication overall.

Bird et al.
Response to Review

Reviewer #1:

The authors have satisfactorily addressed all my concerns and revised the manuscript accordingly. Therefore, this manuscript meets the high-quality standards of Nature Communication.

We are so appreciative of the Reviewer's endorsement of our work for publication in Nature Communications.

Reviewer #2

Authors have addressed all comments raised during peer review; no additional comments.

We thank the Reviewer for their approval of our revision manuscript.

Reviewer #3

The authors have done PK experiments to address my concerns, and I commend them for their further effort. I certainly do not think they need to do further experiments, but I would suggest the following textual modifications:

All three of the reviewers had concerns about the PK and in vivo stability of the new stapled peptides, and while the authors have partly addressed that, key issues remain uncertain. For instance, as they note in their cover letter, the gross amount of compound that is in the general circulation is actually fairly low, probably too low for anti-viral efficacy over the course of a day.

Probably more importantly still, the exposures that they measure--the concentrations in the nares, lung, and the blood--are all gross amounts, not fraction unbound (F_u). It is the latter that is crucial for efficacy, and it will be lower, sometimes much lower, than the gross amount that they measure (e.g., it's not conceivable that the fraction unbound is only 5%, or even less, of the gross amount that they measure).

Given that the blood and even the lung C_{max} values drop from those in the nares (by 100- and 10-fold, respectively), and that the fraction unbound will be lower still, maybe log-orders lower, I recommend that the authors add a caveat to the Discussion, to bring this to the attention of readership. After all, they motivate the study by writing that stapled, lipidated peptides can address what are essentially PK issues. It seems to me that several of these remain unaddressed, or at least uncertain, from their study. Bringing this to the attention of their readers in the Discussion will strengthen the paper, because readers will appreciate their candor.

This is just a recommendation, I leave it to the Editors and the authors discretion as to whether they want to do this. I don't need to see this manuscript again and do support publication overall.

In accordance with the Reviewer's recommendation, we have now supplemented our discussion to clarify that the goal was to deliver our stapled lipopeptide to the nasal mucosa – the initial site of SARS-CoV-2 infection – as the target tissue for therapeutic action. The PK data demonstrate that after a single intranasal dose, nasal tissue levels could be sustained at over 100x the infectivity assay IC90 values for at least 24 h, highlighting durable exposure of lipopeptide at the intended site. Peak lung tissue concentrations were approximately 1/10th of those measured in nasal tissue, whereas plasma exposure was minimal at approximately 1/1000th of that measured in nasal tissue, consistent with our goal of localized topical delivery of RQ-01. We further discuss that expanding the routes of administration to the lung, via nebulization for example, warrants further evaluation as this could be an effective alternative or complementary approach to nasal delivery in order to increase lung exposure. We thank the Reviewer for encouraging us to expand upon these PK-related points in our discussion.